# Research

bioinformatics/evolution/microbiology

metabolic rate, genome size, gene number, cell size, evolution

**Author for correspondence:**
Kazuhiro Takemoto
e-mail: takemoto@bio.kyutech.ac.jp

# Revisiting the hypothesis of an energetic barrier to genome complexity between eukaryotes and prokaryotes

## Katsumi Chiyomaru and Kazuhiro Takemoto

Department of Bioscience and Bioinformatics, Kyushu Institute of Technology, Iizuka, Fukuoka 820-8502, Japan

(iD) KT, 0000-0002-6355-1366

The absence of genome complexity in prokaryotes, being the evolutionary precursors to eukaryotic cells comprising all complex life (the prokaryote–eukaryote divide), is a long-standing question in evolutionary biology. A previous study hypothesized that the divide exists because prokaryotic genome size is constrained by bioenergetics (prokaryotic power per gene or genome being significantly lower than eukaryotic ones). However, this hypothesis was evaluated using a relatively small dataset due to lack of data availability at the time, and is therefore controversial. Accordingly, we constructed a larger dataset of genomes, metabolic rates, cell sizes and ploidy levels to investigate whether an energetic barrier to genome complexity exists between eukaryotes and prokaryotes while statistically controlling for the confounding effects of cell size and phylogenetic signals. Notably, we showed that the differences in bioenergetics between prokaryotes and eukaryotes were less significant than those previously reported. More importantly, we found a limited contribution of power per genome and power per gene to the prokaryote–eukaryote dichotomy. Our findings indicate that the prokaryote–eukaryote divide is hard to explain from the energetic perspective. However, our findings may not entirely discount the traditional hypothesis; in contrast, they indicate the need for more careful examination.

## 1. Introduction

Eukaryotic cells arose from prokaryotes and are comprised of all complex life. Biological complexity (e.g. genome complexity, cellular complexity and multicellularity) is believed to harbour several advantages. Genome size and the number of genes (i.e. genome complexity) increase with environmental variability because organisms need more functional (e.g. metabolic) genes to

adapt to changing environments (e.g. nutrient variability) [1,2]. Cellular complexity may enhance biological robustness [3], and multicellular organisms have evolved sophisticated, higher-level functionality via cooperation among component cells with complementary behaviours [4,5]. However, only some prokaryotes have evolved biological complexity. The large gap between prokaryotes and eukaryotes (the prokaryote–eukaryote divide) is a long-standing mystery in evolutionary biology [6–8].

Lane & Martin focused on genome complexity and hypothesized that the prokaryote–eukaryote divide is due to the prokaryotic genome size being constrained by bioenergetics [6] (Lane–Martin hypothesis). Their report is positioned as a proposal of a hypothesis rather than a data analysis; however, they used the data on genome size, metabolic rate (oxygen consumption rate) and ploidy level of 12 (cellular) eukaryotes and 55 prokaryotes, to demonstrate the validity of their hypothesis. In particular, Lane & Martin showed that the power (i.e. the oxygen available) per gene and power per genome of eukaryotes was significantly larger (approximately 2000-fold) than those of prokaryotes. This indicates the presence of an energetic barrier against genome complexity between prokaryotes and eukaryotes. They concluded that eukaryotes have allowed the expansion of their genome sizes via endosymbiosis, giving rise to mitochondria, which have provided an energetic boost.

However, the Lane–Martin hypothesis is controversial. For example, Lynch [9] pointed out that the increase in genome complexity can be explained through non-adaptive evolutionary processes. Booth and Doolittle argued that eukaryogenesis—the crossing of the deep gulf between prokaryotes and eukaryotes—lacks rigorous evidential and statistical support [10]. The Lane–Martin hypothesis has several limitations. Primarily, the hypothesis was based on a biased evaluation in a limited number of species due to lack of data availability at the time. In addition, the hypothesis was based on the metabolic rate of prokaryotes grown in the presence of various substrates (i.e. under nutrient-rich conditions). The use of such metabolic rates as a measure of power production may not be informative from an evolutionary perspective [11,12].

Moreover, the effects of cell mass were not statistically controlled. Metabolic rate has a strong positive correlation with body mass (cell mass in the case of cellular organisms); in particular, the relationship between metabolic rate and body mass approximately obeys a power law [13–16]. Lynch & Marinov [11,12] investigated a common currency of energy per unit of cell volume and found no energetic difference between eukaryotes and prokaryotes. This finding eliminates the need to invoke an energetics barrier hypothesis to genome complexity; however, it was based on a biased evaluation in a limited number of species. More importantly, the effects of phylogenetic signals were not considered, although the importance of phylogeny in evaluating associations between biological features has been well-established through comparative phylogenetic analyses [17,18]. An opposite conclusion may be derived when considering comparative phylogenetic analysis [19,20].

Therefore, in this study, we revisited the Lane–Martin hypothesis. In particular, a larger dataset of genomes, metabolic rates, ploidy level and cell sizes (masses) was constructed, and the contribution of energetic parameters to prokaryote/eukaryote classification (prokaryote–eukaryote divide) was investigated, while statistically controlling for the potentially confounding effect of cell sizes. Comparative phylogenetic analyses were also performed to evaluate the effects of phylogenetic signals on the contribution of energetic parameters to the prokaryote–eukaryote dichotomy.

# 2. Material and methods

## 2.1. Metabolic rate and cell mass

Based on a previous study [6], we collected data on mass-specific metabolic rates and cell mass of prokaryotes and eukaryotes (protozoa) from the literature [21,22]. Additionally, we used Supporting Datasets S1 (heterotrophic prokaryotes), S2 (heterotrophic protozoa), S7 (cyanobacteria) and S8 (eukaryotic microalgae) from a previous study [16] to obtain additional data on mass-specific metabolic rates (oxygen consumption rate) and cell masses (electronic supplementary material, dataset S1). The units of mass-specific metabolic rates and cell masses were converted to watt per kilogram (W kg$^{-1}$) and picograms (pg), respectively. For a species, multiple values of mass-specific metabolic rates may be available in the dataset. For a comparison with the previous study [6], despite criticism by Lynch & Marinov [11,12], the maximum mass-specific metabolic rates were used to estimate energy supply (electronic supplementary material, dataset S1); specifically, they mainly correspond to mass-specific metabolic rates measured at the exponential or logarithmic growth phase and summit metabolic rates. Moreover, cell mass associated with the metabolic rate was used.

## 2.2. Genome size, gene number and ploidy level

We selected prokaryotic and eukaryotic species whose complete genomes were available in the Kyoto Encyclopedia of Genes and Genomes (KEGG) database [23] and/or National Center for Biotechnology Information (NCBI) database. Haploid genome sizes (bp) and the number of total protein-coding genes (haploid gene number) of these species were obtained from the databases according to species names (electronic supplementary material, dataset S1). When multiple strains for a species were available in the databases, we selected one strain as a representative of the species according to the year in which its genome was first completely determined. Data were not available for some species, and in these instances, the genome size and gene number of a different species within the same genus whose genome was available in the database were used. Specifically, the substitute genome was selected from a different species in the same genus according to the year in which its genome was first completely determined. The data on genome size and gene number were downloaded from the KEGG database on 2 April 2018 and the NCBI database on 12 December 2018, respectively.

Following a previous study [6], we also collected data on the ploidy level (electronic supplementary material, dataset S1). For eukaryotes, ploidy levels were retrieved from the literature according to species names. For prokaryotes, ploidy levels were retrieved from the literature according to species names because some bacteria may be oligoploid and polyploid [24,25]; however, it was assumed that prokaryotes whose ploidy level had not been reported in any previous studies were monoploid. Prokaryotes are generally assumed to be monoploid during slow growth [24]; moreover, the increase in ploidy levels, observed when bacteria grow fast, are transient. For some species, ploidy levels of a different species within the same genus were used because species-specific ploidy levels were unavailable in the dataset. For a given species, multiple values of ploidy levels may be available. In the analyses, the maximum ploidy level was used, which mainly corresponds to the ploidy level observed at the exponential or logarithmic growth phase for each species.

Finally, we obtained a larger dataset of genome, power and cell sizes for 36 eukaryotes and 156 prokaryotes.

## 2.3. Energetic parameters

Following a previous study [6], we used the data on mass-specific metabolic rate ($B_c$), cell mass ($M$), haploid genome size ($G$), haploid gene number ($N_g$) and ploidy level ($P$) to calculate the following energetic parameters: power per cell ($B_c \times M$; fW), power per haploid genome ($1000 \times$ power per cell$/G/P$; pW), power per gene (power per cell$/N_g/P$; fW) and power per genome (power per gene $\times N_g =$ power per cell$/P$; fW). The primary focus was on power per genome and power per gene for comparison with the previous study.

## 2.4. Data analyses

All statistical tests were performed using R software (version 3.6.1; www.R-project.org).

To evaluate the contribution of the energetic parameters and cell size (mass) to the prokaryote–eukaryote dichotomy (or divide), logistic regression analyses were conducted using R software. No biological replicates in the dataset were used in the analyses. The energetic parameters and cell masses were log-transformed. The quantitative variables were normalized to the same scale, with a mean of 0 and a standard deviation of 1, using the scale function in R before the analysis.

To remove the effects of phylogenetic signals from the regression analyses, phylogenetic logistic regression analyses [26,27] were performed using the function *binaryPGLMM* in the R-package *ape* (version 5.3). In this function, s2 is the scaling component of the variance in the model, where s2 = 0 suggests no phylogenetic signal, and a high s2 value implies a strong phylogenetic signal [28]. The phylogenetic tree, required for phylogenetic regression, was constructed using conserved protein-coding genes, following a previous study [29]. The conserved genes were determined based on the KEGG Orthology (KO) database. We selected 17 eukaryotes and 122 prokaryotes available in the KO database and used 12 KO groups conserved in these organisms for phylogenetic tree construction (electronic supplementary material, table S1). The sequences of genes in these groups were downloaded from the KEGG database on January 17, 2019 and were aligned using MUltiple Sequence Comparison by Log-Expectation (MUSCLE; version 3.8.31) [30] with the parameter '-maxiterate 1000' and the resulting alignments were processed using the Gblocks program (version 0.91b) [31] with the default settings to eliminate poorly aligned positions. The processed alignments were concatenated and subjected to phylogenetic analysis. The phylogenetic tree was constructed using Molecular Evolutionary Genetics Analysis (MEGA; version 7) software [32]. We performed model selection based on the Akaike

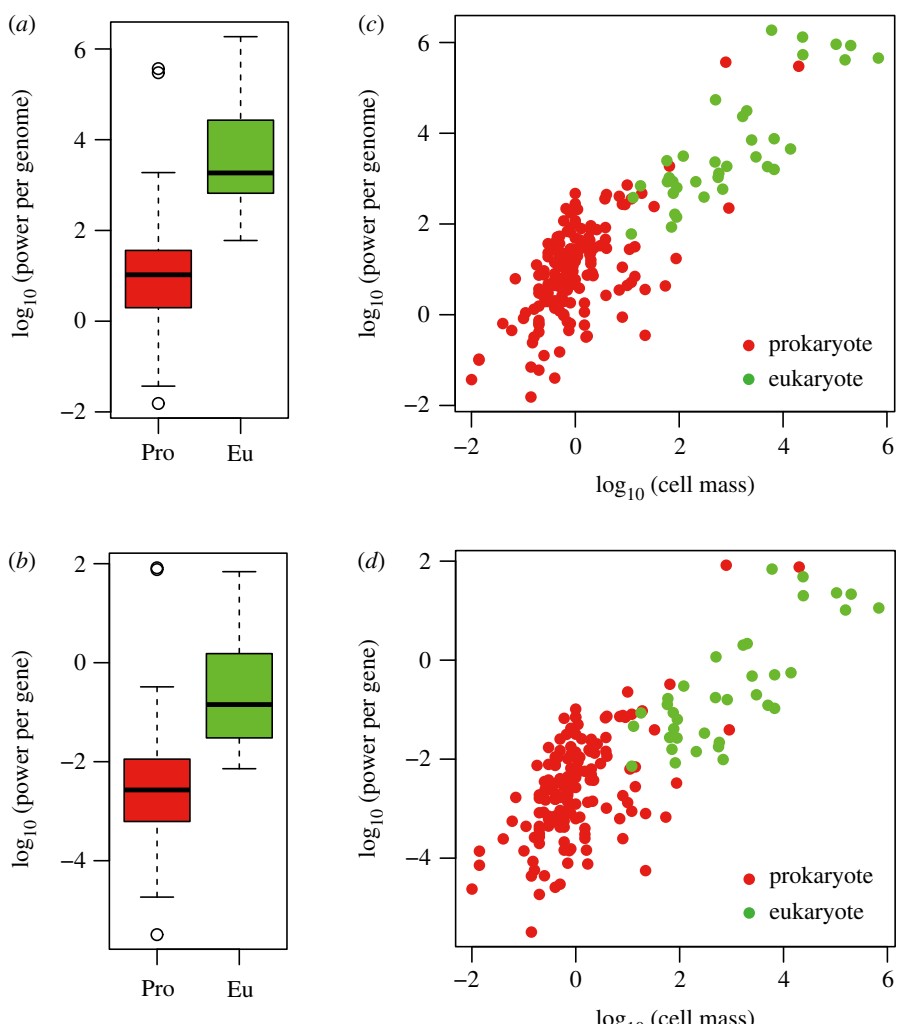

**Figure 1.** Comparison of energetic measures between prokaryotes and eukaryotes. (*a*) The difference in power per genome between prokaryotes (Pro) and eukaryotes (Eu). (*b*) The difference in power per gene between prokaryotes (Pro) and eukaryotes (Eu). (*c*) Scatter plot of power per genome versus cell mass. (*d*) Scatter plot of power per gene and cell mass. Power per genome, power per gene and cell mass were base 10 log-transformed.

information criterion (AIC) value. The substitution LG+G model, the best-fit model, was employed to produce the maximum-likelihood tree (electronic supplementary material, dataset S1 and figure S1).

The contribution (i.e. non-zero estimate) of each explanatory variable to the prokaryote–eukaryote dichotomy was considered significant when the associated *p*-value was less than 0.05. We used the function *R2.pred* in R package *rr2* (version 1.0.2) [28] to calculate the coefficients of determination of the standard and phylogenetic logistic regression models.

## 3. Results

The increased genome complexity in eukaryotes was re-confirmed. The haploid genome sizes (44 Mbp in median) of eukaryotes were larger than that (4 Mbp in median) of prokaryotes ($p < 2.2 \times 10^{-16}$ using the Wilcoxon test). The haploid gene number (14973 in median) of eukaryotes was greater than that (3935 in median) of prokaryotes ($p < 2.2 \times 10^{-16}$ using the Wilcoxon test).

Inspired by the previous study of Lane & Martin [6], we investigated the differences in metabolic power between prokaryotes and eukaryotes and re-confirmed that the metabolic power of eukaryotes was greater than that of prokaryotes. Specifically, the power per cell (4819 fW in median) of eukaryotes was greater than that (1.7 fW in median) of prokaryotes ($p < 2.2 \times 10^{-16}$ using the Wilcoxon test). More importantly, the power per genome (1850 fW in median) of eukaryotes was larger than that (1 fW in median) of prokaryotes (figure 1*a*; $p < 2.2 \times 10^{-16}$ using the Wilcoxon test); in addition,

**Table 1.** Contributions of explanatory variables to the prokaryote–eukaryote dichotomy. The variable 'dichotomy' indicates whether a species is a eukaryote (1) or not (0). s.e. and AIC correspond to the standard error and Akaike information criterion value of the model, respectively. $R^2$ is the coefficient of determination.

| model | variable | estimate | s.e. | $p$-value | AIC | $R^2$ |
|---|---|---|---|---|---|---|
| (a) dichotomy ~ power per genome + cell mass | power per genome | 0.62 | 0.65 | 0.34 | 65.3 | 0.73 |
| | cell mass | 2.88 | 0.71 | $5.1 \times 10^{-5}$ | | |
| (b) dichotomy ~ power per gene + cell mass | power per gene | −0.18 | 0.49 | 0.71 | 67.2 | 0.70 |
| | cell mass | 3.46 | 0.67 | $2.6 \times 10^{-7}$ | | |
| (c) dichotomy ~ power per genome | power per genome | 3.21 | 0.56 | $1.3 \times 10^{-8}$ | 87.3 | 0.62 |
| (d) dichotomy ~ power per gene | power per gene | 2.14 | 0.38 | $1.9 \times 10^{-8}$ | 118.1 | 0.42 |
| (e) dichotomy ~ cell mass | cell mass | 3.33 | 0.57 | $3.9 \times 10^{-9}$ | 65.3 | 0.71 |

the power per haploid genome size ($3.7 \times 10^{-2}$ pW in median) of eukaryotes was larger than that ($2.3 \times 10^{-3}$ pW in median) of prokaryotes ($p = 1.5 \times 10^{-10}$ using the Wilcoxon test). Moreover, the power per gene (0.14 fW in median) of eukaryotes was larger than that (0.0027 fW in median) of prokaryotes (figure 1$b$; $p = 3.1 \times 10^{-15}$ using the Wilcoxon test).

However, differences in cell mass between prokaryotes and eukaryotes were observed; in particular, the cell mass (570 pg median) of eukaryotes was larger than that (0.7 pg in median) of prokaryotes ($p < 2.2 \times 10^{-16}$ using the Wilcoxon test). The distribution of the cell mass of prokaryotes was partially overlapped with that of eukaryotes. For example, the cell mass, power per genome, and power per gene of several prokaryotes (e.g. *Thioploca* and *Trichodesmium* species) almost equalled those of eukaryotes. Moreover, the power per genome and power per gene for prokaryotes appeared to be similar to eukaryotes for a similar cell mass. This indicates that the contributions of power per genome and power per gene in the prokaryote–eukaryote dichotomy depend on cell mass. In particular, the power per genome (figure 1$c$) and power per gene (figure 1$d$) were co-related to cell mass in a linear fashion within and between prokaryotic and eukaryotic groups.

Therefore, a standard logistic regression analysis was performed to statistically control for the effects of cell mass; specifically, multivariate regression models were constructed encompassing an energetic parameter and cell mass. The regression analyses showed that the contributions of power per genome (table 1$a$) and power per gene (table 1$b$) to the prokaryote–eukaryote dichotomy were not statistically significant. Instead, the cell mass was a dominant indicator for the dichotomy (eukaryotic cells are larger than prokaryotic cells). For comparison, single regression analyses (table 1$c$–$e$) were also performed. For power per genome, the AIC values for the single regression model (table 1$c$) were higher than that of the multivariate regression model (table 1$a$). For power per gene, the AIC value of the single model (table 1$d$) was also higher than that of the multivariate model (table 1$b$). These results indicate that cell mass, not power per genome and power per gene, account for the prokaryote–eukaryote dichotomy. In the single regression models, the AIC value of the model for cell mass is the smallest (tie). The result also indicates that cell mass predominately contributes to the prokaryote–eukaryotes dichotomy (table 1$e$).

However, there is a possibility of phylogenetic signals affecting conclusions obtained from the standard regression analyses. Therefore, phylogenetic logistic regression analyses were performed to remove the phylogenetic effects. High s2 values indicated the importance of phylogenetic signals (table 2). The single phylogenetic regression models indicated that the contribution of power per genome (table 2$c$) to the prokaryote–eukaryote dichotomy was less significant, contrary to the results of the standard logistic regression analyses (table 1). The difference in power per gene (table 2$d$) between prokaryotes and eukaryotes was observed; however, this difference was not exceedingly statistically significant. The cell mass of eukaryotes was larger than that of prokaryotes (table 2$e$); however, the contribution of cell mass was not exceedingly significant. Multivariate regression models were also constructed to statistically control for the effect of cell mass. The regression models also showed that the contributions of power per genome (table 2$a$) and power per gene (table 2$b$) to the prokaryote–eukaryote dichotomy were not statistically significant; moreover, cell mass also contributed slightly to the dichotomy.

**Table 2.** Contributions of explanatory variables to the prokaryote–eukaryote dichotomy when removing the effects of phylogenetic signals. The variable 'dichotomy' indicates whether a species is a eukaryote (1) or not (0). s.e. corresponds to the standard error. s2 indicates a phylogenetic signal (see §2.4). Values in brackets are the associated $p$-value. $R^2$ is the coefficient of determination.

| model | variable | estimate | s.e. | $p$-value | s2 | $R^2$ |
|---|---|---|---|---|---|---|
| (a) dichotomy $\sim$ power per genome + cell mass | power per genome | 0.06 | 1.46 | 0.96 | 1.38 $(1.1 \times 10^{-4})$ | 0.98 |
| | cell mass | 1.56 | 1.37 | 0.29 | | |
| (b) dichotomy $\sim$ power per gene + cell mass | power per gene | −0.48 | 1.13 | 0.67 | 1.40 $(1.2 \times 10^{-4})$ | 0.99 |
| | cell mass | 2.09 | 1.40 | 0.14 | | |
| (c) dichotomy $\sim$ power per genome | power per genome | 1.43 | 0.77 | 0.065 | 1.37 $(1.8 \times 10^{-5})$ | 0.99 |
| (d) dichotomy $\sim$ power per gene | power per gene | 1.05 | 0.67 | 0.117 | 1.35 $(2.9 \times 10^{-6})$ | 0.99 |
| (e) dichotomy $\sim$ cell mass | cell mass | 1.61 | 0.76 | 0.035 | 1.35 $(1.1 \times 10^{-4})$ | 0.98 |

## 4. Discussion

These results indicate no difference in power per genome and power per gene between prokaryotes and eukaryotes, which is not consistent with Lane & Martin's conclusion that the prokaryotic genome size is constrained by bioenergetics. The simple comparison tests (figure 1a,b) indicated that the power per genome and median power per gene of eukaryotes were greater than those of prokaryotes; however, the observed differences were artefacts due to no consideration of the effects of cell mass and phylogeny. The result that cell size (mass) showed a linear relationship with power per genome (figure 1c) and power per gene (figure 1d) indicates a lack of difference in power per genome and power per gene between prokaryotes and eukaryotes for similar cell mass. Standard and phylogenetic logistic regression analyses (tables 1a,b and 2a,b) showed no contribution of power per genome and power per gene to the prokaryote–eukaryote dichotomy when statistically controlling for the effect of cell size. Moreover, no difference in power per genome (table 2c) and power per gene between (table 2d) prokaryotes and eukaryotes was observed even if the effect of cell mass was not statistically controlled and disregarding the effects of phylogenetic signals. The results indicate that there is slight difference in power per genome and power per gene at the root of the phylogenetic tree and that a Brownian motion-like evolution could explain the differences in power per genome (figure 1c) and power per gene (figure 1d) observed (i.e. at the leaf level of the tree).

The observed differences in power per genome and power per gene between eukaryotes and prokaryotes were less significant than those previously reported. Specifically, the power per genome of eukaryotes was approximately 1850-fold greater (=1850/1) than that of prokaryotes, although Lane and Martin reported that the power per genome of eukaryotes was approximately 10 000-fold greater (=1143/0.12; see Table 1 in [6]) than that of prokaryotes. Moreover, the power per gene of eukaryotes was approximately 52-fold greater (=0.14/0.0027) than that of prokaryotes, although the previous study reported that the power per gene of eukaryotes was roughly 200-fold greater (=57.15/0.03; see Table 1 in [6]). This discrepancy might be due to differences in the datasets and data analyses between this study and the previous study. In this study, the data on metabolic rate, cell mass and genome were collected from more 36 eukaryotes and 156 prokaryotes, whereas the previous study only considered 12 eukaryotes and 55 prokaryotes. The dataset in this study partly overlaps with the dataset used in the previous study of Lane & Martin [6] because the same literature [21,22] was used. However, we were not able to substantially evaluate how our dataset was different from the dataset used in the previous study by Lane & Martin [6]. We requested the original dataset from one of the authors. However, we were informed that the dataset was currently unavailable; in particular, the author needed to relocate literature sources for genome size, ploidy, metabolic rates, etc., as the original study was much older; as of October 2018. A comparison of datasets may be evaluated in the near future.

The findings of this study are inconsistent with the idea that cells with greater internal complexity impose greater energy supply (i.e. Lane–Martin hypothesis). The findings indicate the prokaryote–eukaryote divide is harder to explain than previously thought; rather, they support the hypothesis of the passive emergence of genome complexity by non-adaptive processes [9,11,12]. As Lynch & Marinov [12] mentioned, the origin of the mitochondrion was not a prerequisite for genome-size expansion, although the origin was a key event in evolutionary history (e.g. the acquisition of eukaryote-specific traits such as the cell cycle, sex, phagocytosis, endomembrane trafficking, the nucleus and multicellularity [6,33]); rather, genome-size expansion passively occurred in species experiencing relatively low efficiency of selection due to small effective population sizes. Koonin [34] also stated that eukaryotic cells emerged at least in part by initial non-adaptive processes made possible due to a strong and prolonged population bottleneck.

The definition of power per genome and power per gene is still a matter of controversy. The conclusion in this study is limited to power per genome and power per gene, as defined in Lane & Martin's original study [6]. As Lynch & Marinov [11,12] pointed out, the use of metabolic rate may not be helpful as a measure of power production, as it may fail to distinguish between the investment in cellular reproduction and that associated with non-growth-related processes (e.g. diversity of cellular functions, ranging from turnover of biomolecules, intracellular transport, control of osmotic balance and membrane potential, nutrient uptake, information processing, and motility). Therefore, more suitable measures are needed for more careful examination. For example, Lynch & Marinov used the number of ATP→ADP turnovers as a common currency of energy and found that the costs of a gene at the DNA, RNA and protein levels declines with cell volume in both bacteria and eukaryotes, relative to the lifetime ATP requirements of a cell. However, Lane & Martin [35] stated that the number of ATP→ADP turnovers is not an alternative measure of the power per gene [6] because it corresponds to energy demand whereas power per gene [6] is considered as energy availability per gene, i.e. supply, not demand. This discrepancy is due to Lane & Martin stating in their original study that power per gene represents the cost of expressing the gene [6]. Lynch & Marinov [36] also pointed out this misleading expression. To avoid this dissonance, a more explicit and easily assessed definition of power per gene may be needed. For example, it may be useful to consider genes in a specific functional category [37]. In this study, the power per gene was used based on the metabolic rate because we aimed to revisit the Lane–Martin hypothesis [6], and the amount of available data on the number of ATP→ADP turnovers was limited. However, as with Lynch & Marinov, our study supports the conclusion that power per gene hardly contributes to the prokaryote–eukaryote divide. In addition, the study of Lynch & Marinov [11] was criticized in terms of elusive data and reproducibility [38]; however, this was caused by the authors' failure to note the citations for these data [39]. This indicates that access to open data is also important for debate in the prokaryote–eukaryote divide.

The current study has several limitations. Only organisms for which complete genome sequences were available were considered in order to accurately estimate power per genome and power per gene and in order to perform the phylogenetic comparative analysis. The findings of this study depend significantly on the quality of genome annotation. Moreover, as previously mentioned [40], there are limitations to the phylogenetic comparative analysis. This type of analysis assumes a Brownian motion-like evolution of biological traits on a phylogenetic tree with accurate branch lengths, which may lead to a misleading conclusion. For example, statistical power decreases when a dataset is reduced in size following phylogenetic corrections [41]. In particular, the dataset used in this study contained only a few samples for eukaryotes. Therefore, the continued sequencing of genomes from a wide range of organisms is important.

The ploidy level is still controversial, although data were collected on ploidy in organisms as much as possible; however, we assumed that prokaryotes whose ploidy level had not been reported in any previous studies were monoploid. This limitation may not pose a problem because bacteria are generally assumed to be monoploid [24] and the increase in ploidy levels, observed when bacteria grow fast, are transient. Moreover, similar tendencies (i.e. limited contributions of power per genome and power per gene to the prokaryote–eukaryote dichotomy) were observed in standard regression analyses even when prokaryotes whose ploidy level had not been reported were removed (electronic supplementary material, table S2). However, the cost of polyploidy should be considered because prokaryotes are believed to require extreme polyploidy to scale up to the eukaryotic size [6,8,35,42]. The prokaryotic ploidy level was correlated with cell mass (Spearman's rank correlation coefficient $r_s = 0.43$ and the associated $p$-value $p = 0.0092$) in the dataset, where the prokaryotes whose ploidy level had not been reported were removed. It may be necessary to consider extreme polyploid bacteria such as *Thiomargarita* and *Epulopiscium*, which have multiple copies of their genome [6,8,35,42], although the dataset in this study included prokaryotes with relatively high ploidy levels (highest value = 218). However, it was

necessary to exclude these species in the data analyses because the parameters required were unavailable and/or ambiguous. For example, accurately annotated genomes are required for calculating power per genome and power per gene and for performing phylogenetic analyses. However, the *Thiomargarita* genome was not complete. The data on the *Epulopiscium* metabolic rate was unavailable. The *Thiomargarita* ploidy level was ambiguous; in particular, it was only retrieved from personal communication [6]. To avoid this limitation, the genomes of more extreme polyploid prokaryotes need to be completed. Moreover, ploidy levels in more organisms need to be identified using real-time polymerase chain reaction (PCR) methods [24,43] because the ploidy level may not be conserved within the same phylogenetic groups, and there may be no obvious correlations between the ploidy levels with primary parameters (e.g. haploid genome size and mode of life) [24].

In conclusion, the findings of this study indicate no energetic barrier to genome complexity between prokaryotes and eukaryotes, contrary to the Lane–Martin hypothesis [6]. Despite the limitations in our data analyses, our findings advance our understanding of the energetics of genome complexity and the prokaryote–eukaryote divide. However, these findings may not entirely discount the traditional hypothesis; instead, they indicate the requirement for a more careful examination using more comprehensive analyses. In particular, this study emphasizes the importance of rigorous evidential and statistical support for debate in the prokaryote–eukaryote divide.

Data accessibility. The datasets supporting this article have been uploaded as electronic supplementary material.

Authors' contributions. K.T. conceived and designed the study. K.C. and K.T. prepared the data. K.C. and K.T. performed data analysis and interpreted the results. K.C. and K.T. drafted the manuscript. All authors have read and approved the final manuscript.

Competing interests. The authors declare no competing interests.

Funding. This study was supported by a Grant-in-Aid for Young Scientists (A) from the Japan Society for the Promotion of Science (grant no. 17H04703).

Acknowledgements. The authors are much obliged to Prof. William F. Martin and Dr Hideto Takami for providing their useful comments on bacterial ploidy. The authors would like to thank Editage (www.editage.com) for English language editing.

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
