## [Reviewer comments · Royal Society Open Science]

Review History

RSOS-191221.R0 (Original submission)

Review form: Reviewer 1

Is the manuscript scientifically sound in its present form?

Yes

Are the interpretations and conclusions justified by the results?

Yes

Is the language acceptable?

Yes

Do you have any ethical concerns with this paper?

No

Have you any concerns about statistical analyses in this paper?

No

Recommendation?

Accept with minor revision (please list in comments)

Comments to the Author(s)

This paper reports on a set of analyses that debunk the claim of Lane and Martin that the origin of the mitochondrion provided a massive boost to the bioenergetic capacity of eukaryotes, spawning in its wake evolutionary diversification of this group. There are two ways of looking at the paper. First, it may be argued that Lynch and Marinov (in two papers) have already provided substantial evidence against the Lane-Martin hypothesis. Second, given Lane and Martin's (especially the latter's) somewhat vitriolic defense of the undefensible, it may be argued that a final nail in the coffin is in order. The authors do take a somewhat different tack than Lynch and Marinov, so this may be viewed as useful. I am not sure the approaches get at the problem as directly as was done by Lynch and Marinov, and in places I think the authors are far too deferential to Lane and Marinov. All in all, this is a difficult call, and I would like to see some modifications, but I don't see anything fundamentally wrong with the paper.

More specific comments:

1. There are a number of places in which the terminology used by the authors is unclear, e.g., the "minimum mass-specific metabolic rate to maintain orderliness of unit live mass". Indeed, in general, I think the authors need to address more directly their whole reliance on metabolic-rate data. What does metabolic rate really mean in evolutionary terms? Ultimately, one would like to know how much different organisms actually put into biomass, and metabolic rate doesn't tell us anything about that – it is simply some measure of how much energy is burned, without providing any direct insights into the benefits. Admittedly, Lane and Martin focused on this kind of currency, and so the authors might be forgiven for going down the same route, but I think it would be best if they addressed the issue of the utility of this measure head on. This is the whole reason that Lynch and Marinov elected to work on cell growth (while not ignoring basal metabolic costs).

2. The authors do quite a bit of waffling about ploidy levels, but I do not really see the huge issue here – after all the cost at the DNA level is quite small to everything else (RNA and protein, as pointed out by Lynch and Marinov), and the issues in prokaryotes are quite different than those in eukaryotes, as in the former ploidy level is transient, and in the latter, there is a wide range of variation in the meaning of the term – from simultaneously segregating multivalents to ancient polyploidization events. To me, this is all a bit of a distraction from the main point, and not likely to be of major relevance.

For similar reasons, I find the use of the currency of "power per gene" as bordering on nonsensical, as it has no obvious relationship to fitness, although I appreciate that the authors are somewhat stuck with this term in their analysis, which attempts to validate the Lane-Martin results. I do think it would still be useful to have a short commentary on the meaningfulness of such an index. Again, I understand that the authors are attempting to replicate the Lane-Martin results, and showing this cannot be done, but I think they are missing the opportunity to weigh in a bit more deeply on the general issues.

3. I find it quite surprising that the authors were unable to get the original data sets that Lane and Martin used in their Nature paper, which seems to be bordering on the unethical from the standpoint of journal requirements. What is particularly concerning here is that Martin has recently attacked Lynch and Marinov in a Biology Direct paper in a somewhat bizarre rant about data unavailability when the data were in fact available. At the very least, the authors here should reference the response by Lynch and Marinov (in Biology Direct as well, which clarifies the situation), and not simply cite reference 17, which will lead the uninformed reader to think there is a legitimate controversy here.

4. In the discussion, the authors state that their results support Lane-Martin's contention that genome size is constrained by bioenergetics. This is a pretty surprising statement. Perhaps the authors are attempting to be nonconfrontational, but I don't see this at all. There is nothing at all in the current paper that suggests that bioenergetics is of any relevance to the prokaryotic –

eukaryotic divide, and repeating this statement is not helpful. It seems that this statement derives from a “power per gene” argument, which is in turn debunked by the authors’ own size-scaling analyses. In the end, the authors even state that they don’t see that their analyses “contradict the traditional hypothesis”. It is unclear what traditional hypothesis is being referred to, but if it is the Lane-Martin hypothesis, this makes little sense, as this hypothesis has hardly been embraced, and a statement like this, which is completely contrary to the paper, will only cause the L-M myth to live on.

In short, the authors present some quite clear-cut results, somewhat redundant to Lynch and Marinov, but done in a different way, but things become muddled at the end when they seem to somehow hold on to the rope for some credence in the Lane-Martin hypothesis even after removing all such credence...

Although I am in favor of publishing this paper, I do think these loose ends need substantial tightening up. The authors have some very clear conclusions, so why leave the door open to debate, especially given the prior evidence against Lane-Martin. I acknowledge that science profits from debate, but the final discussion in this paper is a bit like leaving room for the possibility for intelligent design or for disagreement on global warming.

Review form: Reviewer 2 (Nick Lane)

Is the manuscript scientifically sound in its present form?

No

Are the interpretations and conclusions justified by the results?

No

Is the language acceptable?

Yes

Do you have any ethical concerns with this paper?

No

Have you any concerns about statistical analyses in this paper?

No

Recommendation?

Major revision is needed (please make suggestions in comments)

Comments to the Author(s)

I think it is necessary that I identify myself at the outset on this review: I am Nick Lane, one of the authors of the original Lane & Martin Nature 2010 paper that is under reanalysis here. This means that I have a bias, which it is only fair to recognise. For that reason I do not think that my comments should carry as much weight as the other referees for this paper. However, I should also say upfront that I do not agree with the conclusions of the paper as written. My comments here are sincerely meant to improve the paper. I would hope that they would at least modify the tone and some of the content of a revised MS.

I accepted reviewing this paper because it is obviously important to me. This is a heated field and I dislike the tone of the debate over the last few years, especially between Bill Martin and Mike Lynch, who stands accused of fabricating data. That is as may be. I have other problems with the position of Lynch. The present paper seems to support the views of Lynch and others. If Bill

Martin and I were wrong, I would prefer to be the first person to accept that publicly, hence my agreement to review the paper. Science should aim for the truth, not winning arguments. So I have reviewed this paper with the truth in mind and the intention to be as honest as possible: should I in all honesty put up my hand and say 'we were wrong, our position is now untenable?' The answer on the strength of this analysis is 'no'. I still think our hypothesis is correct, and to my mind this paper misses the point. That is a shame, as the intention of analysing a wider range of data with improved statistical methods is good and potentially important. But the parameters used matter a great deal. Unfortunately the authors have ignored what are in my view the most important parameters.

It seems to me that there has been a systematic error in interpreting what was meant by 'energy per gene' in the original Nature paper, and this error is perpetuated here. It might be that we were not sufficiently clear in that paper, and indeed I have published several other papers that aimed to make at least my own position more clear. However they are rarely cited in comparison. Two of those papers are cited here, but their arguments are not touched upon at all. For example, there are three section headings in the relevant part of Lane 2014 (Cold Spr Harb Persp Biol), each of which is supported by brief argument. They are as follows: 'Scaling of cell volume and genome copy number'; 'The requirement for core bioenergetic genomes'; and 'Endosymbiosis in prokaryotes gives rise to extreme genomic asymmetry'. None of these tenets are even mentioned in the present paper, or previously by Lynch.

The key point for me is this: the original Lane & Martin Nature paper was not primarily an analysis (as described here) but a hypothesis, and published as such. That hypothesis postulates that genes are needed to control respiration, and that only an endosymbiosis can put the right genes in the right place, right next to bioenergetic membranes. The core of the argument is that mitochondria always have a genome, which has been whittled away to nearly nothing, but never lost completely except when ox phos was also lost. We follow John Allen in arguing that these genes are needed to control respiration. That in turn predicts that giant bacteria such as *Epulopiscium* and *Thiomargarita* should have lots of copies of their genome, and indeed they are found to exhibit extreme polyploidy, with tens or hundreds of thousands of copies of their complete genome. When the costs of expressing all that genomic weight is taken into consideration (specifically the costs of gene expression, meaning protein synthesis), then giant bacteria have very much lower energy availability per haploid gene or genome than eukaryotes. Energy per gene always meant the power for gene expression (not gene number, as incorrectly asserted by Lynch).

The present paper does not mention mitochondrial DNA or extreme polyploidy even once (and in all his papers, Lynch has not mentioned mtDNA or extreme polyploidy once). We have said nothing about the efficiency of respiration (as implied here) - we have talked about the ability of cells to SUPPLY enough energy to meet the costs of gene expression when scaling up to eukaryotic volumes. We do not dispute that the energy required for gene expression scales allometrically (or isometrically; it matters little) with the area of bioenergetic membranes. Plainly there is a relationship between the area of bioenergetic membranes and the metabolic rate, and between the number of ribosomes and cell volume and gene expression, as argued by Lynch. There is no reason to think that it should cost more or less to express a gene in bacteria or eukaryotes. Certainly there is a continuum, but there is little if any overlap on this continuum. What does change is supply: the ability of cells to produce enough energy to scale up indefinitely. We contend that to scale up to eukaryotic size requires extreme polyploidy (as with mtDNA in eukaryotes). It is precisely this argument that has been ignored here and in other recent critical literature, despite being explicitly laid out in a reply to Lynch and Marinov (Lane & Martin PNAS 2016), not cited here, and at greater length in Lane J Theoret Biol 2017 (also not cited here). The further point that eukaryotes have a genomic asymmetry, in which tiny mitochondrial genomes support energetically a massively expanded nuclear genome is missing completely.

Below I give some specific comments on the MS.

Abstract: 'the hypothesis is based on a biased evaluation'. Certainly it is based on a small sample size but the hypothesis is based on the reasoning outlined above, and the sample was a small part of that. A large part included reasoning about surface-area-to-volume scaling to control for volume and the costs of extreme polyploidy in giant bacteria. One of the reasons that we had a smaller sample size was precisely that we needed to know the ploidy as we as genome size and cell volume. This has been completely overlooked here.

Abstract: 'Bioenergetically efficient'. We do not claim this (see above). I imagine that prokaryotes are more bioenergetically efficient than eukaryotes. Our claim is that we can explain the absence of what is in fact never seen: we do not see large, eukaryotic-sized prokaryotes with eukaryotic-sized genomes and orders of magnitude more ribosomes etc, combined with internal endomembrane systems, dynamic cytoskeletons, etc. What we see instead is extreme polyploidy with metabolic inertia in the internal volume. None of this is discussed here.

Introduction: 'Lane & Martin used data on genome size and metabolic rate...' this omits to say that we also used ploidy, which is the critical factor as discussed above. So this presents a very partial view already, and I reiterate that this one only part of a hypothesis that was largely reasoned rather than based on data analysis.

'Lynch has pointed out that the increase in genome complexity can be explained by non-adaptive evolutionary processes'. This is neither referenced nor frankly subjected to an iota of thought. Really? So the beauty of the eukaryotic nucleus, with the nuclear pore complexes, elastic lamina, nucleolus, chromosomal packing, meiosis, mitosis, splicing etc etc (I could go on and on) is a result of drift in small populations. It seems to me that is a far weaker claim and should be subject to some serious analysis. Instead it is trotted out without question. But then two sentences later I read that 'the Lane-Martin hypothesis has severe limitations' and was 'based on a biased evaluation of a limited number of species'. See all of the above. I am beginning to wonder if the authors are in some way associated with Lynch because this argument is extremely biased.

'Eukaryotes are no more bioenergetically efficient than eukaryotes'. Again, see above. Lynch missed the point and so do the authors here.

Aims of paper - fails to mention ploidy at all and so completely misses the point of the paper they say they are reanalysing.

Materials and Methods. 'Following a previous study...' again a failure to mention ploidy, which was the key message of the previous study.

'We used minimum mass-specific metabolic rate'. To my mind this is a bizarre decision and undermines the meaning of the whole study. The idea is to test whether there are energetic limitations, and yet the authors use minimum metabolic rate? Limitation means pushing to the limits. Minimum metabolic rate pushes nothing to the limits except possibly how low metabolic rate can get before an organism enters dormancy. It is amazing to me, given this decision, that the authors find any difference at all between eukaryotes and prokaryotes. Dormancy is close to zero metabolic rate regardless of phylogeny. So this study necessarily compresses any differences discussed by Lane & Martin, but this important difference is not highlighted or discussed properly anywhere in the paper - the conclusion that the differences are less extreme than previously reported is presented without this major proviso. Drawing from the same dataset, Lane & Martin had used metabolic rate during the exponential growth phase, which gives some sense of the normal energetic limits of growth. I have to ask myself: is this paper making criticisms of Lane & Martin that I should be worrying about, and the answer at this point is 'no'. This choice of minimum metabolic rates in my view renders the rest of the arguments pointless.

'For prokaryotes, ploidy levels were generally set to one because bacteria are generally assumed to be monoploid during very slow growth.' Again: are the authors doing a reanalysis or are they deliberately setting out to disprove a hypothesis on what I would see as utterly spurious grounds

(again, hidden in the small print of the methods section). 'Some bacteria may be oligoploid or polyploid' hardly gives any indication that the giant bacteria *Epulopiscium* and *Thiomargarita*, which formed the basis of the argument in Lane & Martin, exhibit 'extreme polyploidy' (not our term). These bacteria can have as many as 200,000 copies of their complete genome. But both of these giant bacteria are excluded from the present analysis with no reason given at all - they are simply not mentioned. The concept 'energy per haploid gene' or 'genome' means for each identical copy, so the metabolic rate is divided by the copy number as well as genome size. This is by far the greatest source of disparity between the papers, but is nowhere is this difference even mentioned. At this stage I am wondering if the authors have a deliberate intention to obscure the truth. The highest bacterial ploidy I could discover in the SI was about 20 (I no longer have access to the SI).

Results. Given that the authors use minimal metabolic rates and assume of general ploidy of 1, I am surprised that they still find a difference in energy per gene of around 350-fold.

Correcting for cell mass. We clearly stated in the original paper and in several other papers that the difference between bacteria and eukaryotes primarily reflected the difference in cell volume. We then corrected for cell volume theoretically on the basis of scaling arguments. That exercise was (we stated clearly) simplistic and assumed respiration across the surface of a sphere, but the argument pointed to the requirement for extreme ploidy in scaling up volume over 15,000-fold (the difference in mean volume between the bacteria and the archaea in our study. Our argument was that the larger the bacterium the greater the ploidy ought to be. I stated it even more clearly in Lane CSHP Biol 2014: 'The scaling of polyploid genomes with prokaryotic cell volume is ultimately what prevents both bacteria and archaea from attaining eukaryotic complexity (but has been ignored by some; Lynch and Marinov, 2015, 2017). Yet the logistic regression analyses performed here to control for the effects of cell mass did not include ploidy - the one factor that we explicitly predicted should correlate with cell mass. While it is true that there is not much reliable data on ploidy in bacteria, the authors have not even discussed the issue.

Discussion. 'We used the minimum mass-specific metabolic rates to accurately estimate cellular maintenance costs'. I reiterate my point from above: cellular maintenance costs have nothing whatsoever to do with the question at hand.

'We were informed the dataset was presently unavailable'. I had responded to the email from Kazuhiro Takemoto to say: 'Thanks for your interest. I'll try to dig it out - I don't recall a dataset as such for that table, more simply literature sources for genome size, ploidy, metabolic rates etc. I'll need to relocate them as it was quite a long time ago now. And I'm mostly travelling in the next 2-3 weeks but I'll do my best to get it to you ASAP.' I regret that I failed to get back to Kazuhiro after my travels, but had been hard pressed for a long period. I would have had to mine my original sources in much the same way that they would: all our data were sourced from the literature and referenced appropriately. I do not think that what the text says about this is strictly accurate: it implies that we were deliberately concealing data. Nor should I be acknowledged for 'useful comments on data availability'. That implies I am complicit somehow in this paper, which I am not.

'Lack of difference in power per genome and power per gene between prokaryotes and eukaryotes of similar mass.' The Lane-Martin prediction is that there should be no difference between small eukaryotes and prokaryotes (where ploidy is 1), but the larger the bacteria, the greater the ploidy and the greater the difference should be - so long as ploidy is taken into consideration, and so long as metabolic rate during exponential growth rather than minimum maintenance are taken into consideration. See above.

'Our findings are inconsistent with the idea that cells with greater internal complexity impose greater energy demands'. This is absurd. Of course large complex cells have greater energy demands. The argument here should be that large complex bacteria would also have greater energy demands. But the problem is that there are NO large, complex bacteria, so this contention

cannot be tested through the analysis of any databases. This lack of overlap is unclear in the present paper (but is clear in Lynch papers) because minimum maintenance metabolic rates were used here, without considering ploidy. These difference would surely have been far more obvious had the analysis used exponential growth rates and extreme polyploidy.

'Eukaryotes are not bioenergetically more efficient than prokaryotes'. We never said they were. See above.

The whole section on the similarity of mean MINIMUM mass-specific metabolic rates is not relevant to the question in my view as it minimises the differences between cells as well as ploidy levels.

'We only considered organisms for which complete genome sequences were available.' Another strange choice. The only reason to use a complete genome sequence is to confirm the number of open reading frames. But if an incompletely sequenced genome is thought to have 4000 open reading frames, it is unlikely that the margin for error would be more than say 20% - perhaps there might be 5000 or 3000 open reading frames, but the margin of error is small. But if the actual ploidy is 2, then the estimate of energy per genome is out by 2-fold. In the case of giant bacteria, it would be out by many thousands-fold. So the use of full genome sequence data is irrelevant, and in fact excludes giant bacteria that have not been fully sequenced yet, but whose ploidy has previously been robustly estimated.

The only comment on ploidy in the discussion is the following: 'We assumed that prokaryotes whose ploidy had been reported in previous studies were monoploid. This limitation may pose little problem because bacteria are generally assumed to be monoploid'. This comment says it all really. I can only think that the authors did not bother to read any of our papers on this matter, as our whole argument is based on extreme ploidy in giant bacteria. How can this study claim to be a reanalysis when it does not even begin to ask the same questions?

I honestly came to this paper with the intent to read it carefully and hopefully learn something; and if the analysis challenged what I had thought to be true, then I should modify my opinion. I hope that this review makes it abundantly clear why I do not think this analysis challenges in any meaningful way the Lane-Martin hypothesis. I cannot recommend the publication of this paper in its current form.

Yours sincerely
Nick Lane

Decision letter (RSOS-191221.R0)

28-Sep-2019

Dear Dr Takemoto:

Manuscript ID RSOS-191221 entitled "Revisiting the energetic barrier to genome complexity between eukaryotes and prokaryotes" which you submitted to Royal Society Open Science, has been reviewed. The comments from reviewers are included at the bottom of this letter.

As discussed by the Associate Editor there are very polarised views as to your work and its conclusions. However, it is important that you take into account the thoughts of the reviewers, particularly the additional points made with respect to the comments of Nick Lane, which should be addressed. Thus the manuscript has been rejected in its current form. However, we very much

welcome a new revised manuscript which takes into consideration these comments. The revised paper will be handled as a new submission and will be re-reviewed.

Please note that resubmitting your manuscript does not guarantee eventual acceptance, and that your resubmission will be subject to peer review before a decision is made.

Your resubmitted manuscript should be submitted by 27-Mar-2020. If you are unable to submit by this date please contact the Editorial Office.

on behalf of Dr Rees Kassen (Associate Editor) and Steve Brown (Subject Editor)
openscience@royalsociety.org

Associate Editor Comments to Author (Dr Rees Kassen):

Associate Editor: 1

Comments to the Author:

Your paper has now been seen by two reviewers, one of whom (Nick Lane) has waived their right to anonymity, as you will see. On the basis of these reviews, I am recommending the paper be rejected and resubmission be allowed and, indeed, encouraged.

The reviewers present strongly contrasting views of the manuscript. This is perhaps not surprising given the controversy - and at times vitriol - surrounding the hypothesis you are testing. In crafting my recommendation, I sought to take these highly polarized views into account.

The key observation demanding explanation is that eukaryotes are so much more complex, in terms of genome size and architecture as well as other features (cell morphology, multicellularity, etc...), than prokaryotes. The controversy in the literature is over why. Lane and Martin (2010) argued that complexity evolved as a consequence of the endosymbiosis that gave us mitochondria, which increased the supply of energy available to the cell and so permitted complexity to evolve. Lynch and Marinov (2015) argued, in response, that complexity is a consequence of weak selection against excess DNA driven by an increase in eukaryotic cell size, and so decrease in effective population size. The current paper seeks to confront the Lane and Martin hypothesis directly with more data and a more sophisticated analysis. It does not take on directly the Lynch and Marinov hypothesis.

The results are interesting - the apparent increase in energy supply posited by Lane and Martin appears to be much smaller once cell size and phylogeny are taken into account, weakening the energy-supply hypothesis. Nick Lane has, correctly I think, pointed out that at least two features of the current data set must be incorporated to provide a more compelling test: data on ploidy

levels and some measure of exponential growth rate (rather than minimum metabolic rate alone). I concur, if only because doing so would provide a more direct comparison with the original work. A revised ms should include responses to all other criticisms as well, of course.

I therefore invite the authors to perform these analyses and resubmit their results. At that time I would seek to identify at least one, if not two, additional reviewers who are less intellectually (and perhaps emotionally) invested in the results.

Reviewers' Comments to Author:

Reviewer: 1

Comments to the Author(s)

This paper reports on a set of analyses that debunk the claim of Lane and Martin that the origin of the mitochondrion provided a massive boost to the bioenergetic capacity of eukaryotes, spawning in its wake evolutionary diversification of this group. There are two ways of looking at the paper. First, it may be argued that Lynch and Marinov (in two papers) have already provided substantial evidence against the Lane-Martin hypothesis. Second, given Lane and Martin's (especially the latter's) somewhat vitriolic defense of the undefensible, it may be argued that a final nail in the coffin is in order. The authors do take a somewhat different tack than Lynch and Marinov, so this may be viewed as useful. I am not sure the approaches get at the problem as directly as was done by Lynch and Marinov, and in places I think the authors are far too deferential to Lane and Marinov. All in all, this is a difficult call, and I would like to see some modifications, but I don't see anything fundamentally wrong with the paper.

More specific comments:

1. There are a number of places in which the terminology used by the authors is unclear, e.g., the "minimum mass-specific metabolic rate to maintain orderliness of unit live mass". Indeed, in general, I think the authors need to address more directly their whole reliance on metabolic-rate data. What does metabolic rate really mean in evolutionary terms? Ultimately, one would like to know how much different organisms actually put into biomass, and metabolic rate doesn't tell us anything about that – it is simply some measure of how much energy is burned, without providing any direct insights into the benefits. Admittedly, Lane and Martin focused on this kind of currency, and so the authors might be forgiven for going down the same route, but I think it would be best if they addressed the issue of the utility of this measure head on. This is the whole reason that Lynch and Marinov elected to work on cell growth (while not ignoring basal metabolic costs).

2. The authors do quite a bit of waffling about ploidy levels, but I do not really see the huge issue here – after all the cost at the DNA level is quite small to everything else (RNA and protein, as pointed out by Lynch and Marinov), and the issues in prokaryotes are quite different than those in eukaryotes, as in the former ploidy level is transient, and in the latter, there is a wide range of variation in the meaning of the term – from simultaneously segregating multivalents to ancient polyploidization events. To me, this is all a bit of a distraction from the main point, and not likely to be of major relevance.

For similar reasons, I find the use of the currency of "power per gene" as bordering on nonsensical, as it has no obvious relationship to fitness, although I appreciate that the authors are somewhat stuck with this term in their analysis, which attempts to validate the Lane-Martin results. I do think it would still be useful to have a short commentary on the meaningfulness of such an index. Again, I understand that the authors are attempting to replicate the Lane-Martin results, and showing this cannot be done, but I think they are missing the opportunity to weigh in a bit more deeply on the general issues.

3. I find it quite surprising that the authors were unable to get the original data sets that Lane and

Martin used in their Nature paper, which seems to be bordering on the unethical from the standpoint of journal requirements. What is particularly concerning here is that Martin has recently attacked Lynch and Marinov in a Biology Direct paper in a somewhat bizarre rant about data unavailability when the data were in fact available. At the very least, the authors here should reference the response by Lynch and Marinov (in Biology Direct as well, which clarifies the situation), and not simply cite reference 17, which will lead the uninformed reader to think there is a legitimate controversy here.

4. In the discussion, the authors state that their results support Lane-Martin's contention that genome size is constrained by bioenergetics. This is a pretty surprising statement. Perhaps the authors are attempting to be nonconfrontational, but I don't see this at all. There is nothing at all in the current paper that suggests that bioenergetics is of any relevance to the prokaryotic - eukaryotic divide, and repeating this statement is not helpful. It seems that this statement derives from a "power per gene" argument, which is in turn debunked by the authors' own size-scaling analyses. In the end, the authors even state that they don't see that their analyses "contradict the traditional hypothesis". It is unclear what traditional hypothesis is being referred to, but if it is the Lane-Martin hypothesis, this makes little sense, as this hypothesis has hardly been embraced, and a statement like this, which is completely contrary to the paper, will only cause the L-M myth to live on.

In short, the authors present some quite clear-cut results, somewhat redundant to Lynch and Marinov, but done in a different way, but things become muddled at the end when they seem to somehow hold on to the rope for some credence in the Lane-Martin hypothesis even after removing all such credence...

Although I am in favor of publishing this paper, I do think these loose ends need substantial tightening up. The authors have some very clear conclusions, so why leave the door open to debate, especially given the prior evidence against Lane-Martin. I acknowledge that science profits from debate, but the final discussion in this paper is a bit like leaving room for the possibility for intelligent design or for disagreement on global warming.

Reviewer: 2

Comments to the Author(s)

I think it is necessary that I identify myself at the outset on this review: I am Nick Lane, one of the authors of the original Lane & Martin Nature 2010 paper that is under reanalysis here. This means that I have a bias, which it is only fair to recognise. For that reason I do not think that my comments should carry as much weight as the other referees for this paper. However, I should also say upfront that I do not agree with the conclusions of the paper as written. My comments here are sincerely meant to improve the paper. I would hope that they would at least modify the tone and some of the content of a revised MS.

I accepted reviewing this paper because it is obviously important to me. This is a heated field and I dislike the tone of the debate over the last few years, especially between Bill Martin and Mike Lynch, who stands accused of fabricating data. That is as may be. I have other problems with the position of Lynch. The present paper seems to support the views of Lynch and others. If Bill Martin and I were wrong, I would prefer to be the first person to accept that publicly, hence my agreement to review the paper. Science should aim for the truth, not winning arguments. So I have reviewed this paper with the truth in mind and the intention to be as honest as possible: should I in all honesty put up my hand and say 'we were wrong, our position is now untenable?' The answer on the strength of this analysis is 'no'. I still think our hypothesis is correct, and to my mind this paper misses the point. That is a shame, as the intention of analysing a wider range of data with improved statistical methods is good and potentially important. But the parameters used matter a great deal. Unfortunately the authors have ignored what are in my view the most important parameters.

It seems to me that there has been a systematic error in interpreting what was meant by 'energy per gene' in the original Nature paper, and this error is perpetuated here. It might be that we were not sufficiently clear in that paper, and indeed I have published several other papers that aimed to make at least my own position more clear. However they are rarely cited in comparison. Two of those papers are cited here, but their arguments are not touched upon at all. For example, there are three section headings in the relevant part of Lane 2014 (Cold Spr Harb Persp Biol), each of which is supported by brief argument. They are as follows: 'Scaling of cell volume and genome copy number'; 'The requirement for core bioenergetic genomes'; and 'Endosymbiosis in prokaryotes gives rise to extreme genomic asymmetry'. None of these tenets are even mentioned in the present paper, or previously by Lynch.

The key point for me is this: the original Lane & Martin Nature paper was not primarily an analysis (as described here) but a hypothesis, and published as such. That hypothesis postulates that genes are needed to control respiration, and that only an endosymbiosis can put the right genes in the right place, right next to bioenergetic membranes. The core of the argument is that mitochondria always have a genome, which has been whittled away to nearly nothing, but never lost completely except when ox phos was also lost. We follow John Allen in arguing that these genes are needed to control respiration. That in turn predicts that giant bacteria such as *Epulopiscium* and *Thiomargarita* should have lots of copies of their genome, and indeed they are found to exhibit extreme polyploidy, with tens or hundreds of thousands of copies of their complete genome. When the costs of expressing all that genomic weight is taken into consideration (specifically the costs of gene expression, meaning protein synthesis), then giant bacteria have very much lower energy availability per haploid gene or genome than eukaryotes. Energy per gene always meant the power for gene expression (not gene number, as incorrectly asserted by Lynch).

The present paper does not mention mitochondrial DNA or extreme polyploidy even once (and in all his papers, Lynch has not mentioned mtDNA or extreme polyploidy once). We have said nothing about the efficiency of respiration (as implied here) - we have talked about the ability of cells to SUPPLY enough energy to meet the costs of gene expression when scaling up to eukaryotic volumes. We do not dispute that the energy required for gene expression scales allometrically (or isometrically; it matters little) with the area of bioenergetic membranes. Plainly there is a relationship between the area of bioenergetic membranes and the metabolic rate, and between the number of ribosomes and cell volume and gene expression, as argued by Lynch. There is no reason to think that it should cost more or less to express a gene in bacteria or eukaryotes. Certainly there is a continuum, but there is little if any overlap on this continuum. What does change is supply: the ability of cells to produce enough energy to scale up indefinitely. We contend that to scale up to eukaryotic size requires extreme polyploidy (as with mtDNA in eukaryotes). It is precisely this argument that has been ignored here and in other recent critical literature, despite being explicitly laid out in a reply to Lynch and Marinov (Lane & Martin PNAS 2016), not cited here, and at greater length in Lane J Theoret Biol 2017 (also not cited here). The further point that eukaryotes have a genomic asymmetry, in which tiny mitochondrial genomes support energetically a massively expanded nuclear genome is missing completely.

Below I give some specific comments on the MS.

Abstract: 'the hypothesis is based on a biased evaluation'. Certainly it is based on a small sample size but the hypothesis is based on the reasoning outlined above, and the sample was a small part of that. A large part included reasoning about surface-area-to-volume scaling to control for volume and the costs of extreme polyploidy in giant bacteria. One of the reasons that we had a smaller sample size was precisely that we needed to know the ploidy as well as genome size and cell volume. This has been completely overlooked here.

Abstract: 'Bioenergetically efficient'. We do not claim this (see above). I imagine that prokaryotes are more bioenergetically efficient than eukaryotes. Our claim is that we can explain the absence

of what is in fact never seen: we do not see large, eukaryotic-sized prokaryotes with eukaryotic-sized genomes and orders of magnitude more ribosomes etc, combined with internal endomembrane systems, dynamic cytoskeletons, etc. What we see instead is extreme polyploidy with metabolic inertia in the internal volume. None of this is discussed here.

Introduction: 'Lane & Martin used data on genome size and metabolic rate...' this omits to say that we also used ploidy, which is the critical factor as discussed above. So this presents a very partial view already, and I reiterate that this one only part of a hypothesis that was largely reasoned rather than based on data analysis.

'Lynch has pointed out that the increase in genome complexity can be explained by non-adaptive evolutionary processes'. This is neither referenced nor frankly subjected to an iota of thought. Really? So the beauty of the eukaryotic nucleus, with the nuclear pore complexes, elastic lamina, nucleolus, chromosomal packing, meiosis, mitosis, splicing etc etc (I could go on and on) is a result of drift in small populations. It seems to me that is a far weaker claim and should be subject to some serious analysis. Instead it is trotted out without question. But then two sentences later I read that 'the Lane-Martin hypothesis has severe limitations' and was 'based on a biased evaluation of a limited number of species'. See all of the above. I am beginning to wonder if the authors are in some way associated with Lynch because this argument is extremely biased.

'Eukaryotes are no more bioenergetically efficient than eukaryotes'. Again, see above. Lynch missed the point and so do the authors here.

Aims of paper - fails to mention ploidy at all and so completely misses the point of the paper they say they are reanalysing.

Materials and Methods. 'Following a previous study...' again a failure to mention ploidy, which was the key message of the previous study.

'We used minimum mass-specific metabolic rate'. To my mind this is a bizarre decision and undermines the meaning of the whole study. The idea is to test whether there are energetic limitations, and yet the authors use minimum metabolic rate? Limitation means pushing to the limits. Minimum metabolic rate pushes nothing to the limits except possibly how low metabolic rate can get before an organism enters dormancy. It is amazing to me, given this decision, that the authors find any difference at all between eukaryotes and prokaryotes. Dormancy is close to zero metabolic rate regardless of phylogeny. So this study necessarily compresses any differences discussed by Lane & Martin, but this important difference is not highlighted or discussed properly anywhere in the paper - the conclusion that the differences are less extreme than previously reported is presented without this major proviso. Drawing from the same dataset, Lane & Martin had used metabolic rate during the exponential growth phase, which gives some sense of the normal energetic limits of growth. I have to ask myself: is this paper making criticisms of Lane & Martin that I should be worrying about, and the answer at this point is 'no'. This choice of minimum metabolic rates in my view renders the rest of the arguments pointless.

'For prokaryotes, ploidy levels were generally set to one because bacteria are generally assumed to be monoploid during very slow growth.' Again: are the authors doing a reanalysis or are they deliberately setting out to disprove a hypothesis on what I would see as utterly spurious grounds (again, hidden in the small print of the methods section). 'Some bacteria may be oligoploid or polyploid' hardly gives any indication that the giant bacteria *Epulopiscium* and *Thiomargarita*, which formed the basis of the argument in Lane & Martin, exhibit 'extreme polyploidy' (not our term). These bacteria can have as many as 200,000 copies of their complete genome. But both of these giant bacteria are excluded from the present analysis with no reason given at all - they are simply not mentioned. The concept 'energy per haploid gene' or 'genome' means for each identical copy, so the metabolic rate is divided by the copy number as well as genome size. This is by far the greatest source of disparity between the papers, but is nowhere is this difference even mentioned. At this stage I am wondering if the authors have a deliberate intention to obscure the

truth. The highest bacterial ploidy I could discover in the SI was about 20 (I no longer have access to the SI).

Results. Given that the authors use minimal metabolic rates and assume of general ploidy of 1, I am surprised that they still find a difference in energy per gene of around 350-fold.

Correcting for cell mass. We clearly stated in the original paper and in several other papers that the difference between bacteria and eukaryotes primarily reflected the difference in cell volume. We then corrected for cell volume theoretically on the basis of scaling arguments. That exercise was (we stated clearly) simplistic and assumed respiration across the surface of a sphere, but the argument pointed to the requirement for extreme ploidy in scaling up volume over 15,000-fold (the difference in mean volume between the bacteria and the archaea in our study). Our argument was that the larger the bacterium the greater the ploidy ought to be. I stated it even more clearly in Lane CSHP Biol 2014: 'The scaling of polyploid genomes with prokaryotic cell volume is ultimately what prevents both bacteria and archaea from attaining eukaryotic complexity (but has been ignored by some; Lynch and Marinov, 2015, 2017). Yet the logistic regression analyses performed here to control for the effects of cell mass did not include ploidy - the one factor that we explicitly predicted should correlate with cell mass. While it is true that there is not much reliable data on ploidy in bacteria, the authors have not even discussed the issue.

Discussion. 'We used the minimum mass-specific metabolic rates to accurately estimate cellular maintenance costs'. I reiterate my point from above: cellular maintenance costs have nothing whatsoever to do with the question at hand.

'We were informed the dataset was presently unavailable'. I had responded to the email from Kazuhiro Takemoto to say: 'Thanks for your interest. I'll try to dig it out - I don't recall a dataset as such for that table, more simply literature sources for genome size, ploidy, metabolic rates etc. I'll need to relocate them as it was quite a long time ago now. And I'm mostly travelling in the next 2-3 weeks but I'll do my best to get it to you ASAP.' I regret that I failed to get back to Kazuhiro after my travels, but had been hard pressed for a long period. I would have had to mine my original sources in much the same way that they would: all our data were sourced from the literature and referenced appropriately. I do not think that what the text says about this is strictly accurate: it implies that we were deliberately concealing data. Nor should I be acknowledged for 'useful comments on data availability'. That implies I am complicit somehow in this paper, which I am not.

'Lack of difference in power per genome and power per gene between prokaryotes and eukaryotes of similar mass.' The Lane-Martin prediction is that there should be no difference between small eukaryotes and prokaryotes (where ploidy is 1), but the larger the bacteria, the greater the ploidy and the greater the difference should be - so long as ploidy is taken into consideration, and so long as metabolic rate during exponential growth rather than minimum maintenance are taken into consideration. See above.

'Our findings are inconsistent with the idea that cells with greater internal complexity impose greater energy demands'. This is absurd. Of course large complex cells have greater energy demands. The argument here should be that large complex bacteria would also have greater energy demands. But the problem is that there are NO large, complex bacteria, so this contention cannot be tested through the analysis of any databases. This lack of overlap is unclear in the present paper (but is clear in Lynch papers) because minimum maintenance metabolic rates were used here, without considering ploidy. These difference would surely have been far more obvious had the analysis used exponential growth rates and extreme polyploidy.

'Eukaryotes are not bioenergetically more efficient than prokaryotes'. We never said they were. See above.

The whole section on the similarity of mean MINIMUM mass-specific metabolic rates is not

relevant to the question in my view as it minimises the differences between cells as well as ploidy levels.

'We only considered organisms for which complete genome sequences were available.' Another strange choice. The only reason to use a complete genome sequence is to confirm the number of open reading frames. But if an incompletely sequenced genome is thought to have 4000 open reading frames, it is unlikely that the margin for error would be more than say 20% - perhaps there might be 5000 or 3000 open reading frames, but the margin of error is small. But if the actual ploidy is 2, then the estimate of energy per genome is out by 2-fold. In the case of giant bacteria, it would be out by many thousands-fold. So the use of full genome sequence data is irrelevant, and in fact excludes giant bacteria that have not been fully sequenced yet, but whose ploidy has previously been robustly estimated.

The only comment on ploidy in the discussion is the following: 'We assumed that prokaryotes whose ploidy had been reported in previous studies were monoploid. This limitation may pose little problem because bacteria are generally assumed to be monoploid'. This comment says it all really. I can only think that the authors did not bother to read any of our papers on this matter, as our whole argument is based on extreme ploidy in giant bacteria. How can this study claim to be a reanalysis when it does not even begin to ask the same questions?

I honestly came to this paper with the intent to read it carefully and hopefully learn something; and if the analysis challenged what I had thought to be true, then I should modify my opinion. I hope that this review makes it abundantly clear why I do not think this analysis challenges in any meaningful way the Lane-Martin hypothesis. I cannot recommend the publication of this paper in its current form.

Yours sincerely
Nick Lane

Author's Response to Decision Letter for (RSOS-191221.R0)

See Appendix A.

RSOS-191859.R0

Review form: Reviewer 3

Is the manuscript scientifically sound in its present form?

No

Are the interpretations and conclusions justified by the results?

No

Is the language acceptable?

No

Do you have any ethical concerns with this paper?

Yes

Have you any concerns about statistical analyses in this paper?

No

Recommendation?

Reject

Comments to the Author(s)

Comments on RSOS 191859

Chiyomaru and Takemoto (C&T) revisit the energetics of genome complexity by gathering more data. In the foreground of the file on this paper stands a debate between two L&M teams Lynch&Marionov 2015 (LM15) and Lane&Martin 2010 (LM10). The peer review on the paper has been intense with R1 for the Lynch side talking about "nails in coffins", "nonsensical", a "bizzare rant" versus R2 for the Lane side pointing out in very level headed words that the T&C paper has some issues and that the Lynch side still does not get the point.

The favourable referee R1 weighing in for Lynch says that C&T is "redundant" with earlier work, and that the case for publication is weak ("a difficult call") and that the conclusions are "muddled" because C&T cannot find that L&M10 are really wrong in any way, findings that R1 requests to be restated so as to support a different conclusions. The critical referee R2 has identified fundamental problems with the paper that revisions and new analyses have not resolved.

There has been enough written on this paper, a decision is needed. Referees are supposed to advise Editors. My recommendation to the editor is to decline. The reason is that this referee looked at the RSOS guidelines which state:

Papers should only be recommended for publication if they meet these [five] criteria:

- [1] - Original research which has not been published elsewhere.
- [2] - Submissions should sufficiently advance scientific knowledge. Negative findings, meta-analyses and studies testing reproducibility of significant work are also encouraged. Repeated experiments will only be considered if they provide a meaningful contribution to the literature. Derivative work will not be considered.
- [3] - Conclusions are supported by the data; data supporting the findings of the paper are publicly available.
- [4] - Compliance with appropriate ethical standards.
- [5] - Experimental protocols/procedure and statistical analysis are performed to a high technical standard and are both methodologically and scientifically sound.

Criterion [1] appears to be met.

Criterion [2], the referees' long comments and the debate between referees in the present file show how incremental the advance is, if there is one. C&T find that L&M10 is reproducible, the first 2 paragraphs of the results state this clearly. Then comes a discussion about cell mass, which is volume because unicellular forms cells have roughly the same density and are 80% water. It is known that some prokaryotes are large, as both L&Ms point out. On lines 200-214 T&C report analyses indicating the eukaryote cells are larger than prokaryote cells (l. 206); that is decidedly not an advance. On lines 215-229 T&C then show that cell mass (volume) also has a phylogenetic component. The result is Table 2 where the reader is challenged to find a strong statistical signal - the values are all very similar.

Criterion [3], the conclusions of L&M10 are supported by the present data, as R1 recurrently points out asking that it not be mentioned, not the conclusions of C&T that L&M15 are supported. The data appears to be available.

Criterion [4], compliance with appropriate ethical standards, there are some accusations about data availability in the text that are hardly acceptable for a scientific paper, in my view. That is not good from an ethics standpoint. The issue on the debate ref 37 vs 38 is 37 having been rejected (37 states so in the text) by the journal that published ref. 11.

Criterion [5], this referee is not an expert in statistics.

The discussion starts out by saying that (L231) "These results indicate no difference in power per genome and power per gene between prokaryotes and eukaryotes, which is not consistent with Lane and Martin's conclusion that the prokaryotic genome size is constrained by bioenergetics". From there the discussion gets very, very dense, missing the bigger picture presented in L&M10. I think the broad appeal of L&M10 is to the community of physiologists and cell biologists. Eukaryotes respire in mitochondria, prokaryotes respire at the plasma membrane, eukaryotes are complex, prokaryotes are not, eukaryotes have meiosis, prokaryotes do not.

On the bottom line, as this reader/referee sees it, C&T say that mitochondria had no role in evolution. L&M10 are saying that symbiosis might be an important force in evolution that can create novelty, while L&M15 are saying that that all evolution needs in order to create novelty is variation in population size. C&T report data that R1 finds redundant and R2 finds to be entirely missing the point.

I think that R2 has the better arguments, hence my recommendation above. I really like symbiosis in evolution, hence I like L&M10. Martin published a good paper entitled "Symbiogenesis, gradualism and mitochondrial energy in eukaryote evolution" on this debate in an obscure but readily accessible open access journal. He pointed out that mitochondria produce about 95% of the ATP in a respiring cell, but comprise only 10% of the cell's volume, releasing constraints on the remaining 90% of the cell (mostly cytosol) to evolve other functions, structural proteins rather than enzymes, while leaving everything under that control of a handful of more genes whose expression not copy number is increased. How can that be irrelevant for evolution? Discussing that would help to explain what the actual issue is here, because it was the role of mitochondria in the origin of complexity of eukaryotic cells that was at the heart of the L&M10 paper. Especially the Discussion is not focussed on science, it is about who did what to whom, which is immaterial for understanding the role of mitochondria in eukaryote origin. The "debate" as presented in the C&T paper is hardly enlightening, as both R1 and R2 conclude, albeit for very different reasons.

Review form: Reviewer 4

Is the manuscript scientifically sound in its present form?

Yes

Are the interpretations and conclusions justified by the results?

Yes

Is the language acceptable?

Yes

Do you have any ethical concerns with this paper?

No

Have you any concerns about statistical analyses in this paper?

No

Recommendation?

Accept as is

Comments to the Author(s)

I've checked all the boxes because they are required, but I really have not had the time to reason my way through the arguments here, in which I am not in any case an expert. My concerns about Lane and Martin 2010 are nevertheless emphasized by authors' Fig. 1, which shows overlap between prokaryotes and eukaryotes, however measured. This seems to me to make the only point that needs to be made: we might easily imagine a large-genomed, high-energy prokaryote "evolving into" a small-genomed, low-energy eukaryote. Indeed that's what most sensible folks would imagine - not that something like the average modern prokaryote "evolved into" something like the average modern eukaryote. Thus comparisons of averages seem to me beside the point. Moreover, there are eukaryotes without mitochondria that have most of the bells and whistles of other eukaryotes without the mitochondrion-derived energy. So such a scenario is possible.

Review form: Reviewer 5**Is the manuscript scientifically sound in its present form?**

Yes

Are the interpretations and conclusions justified by the results?

Yes

Is the language acceptable?

Yes

Do you have any ethical concerns with this paper?

No

Have you any concerns about statistical analyses in this paper?

No

Recommendation?

Accept as is

Comments to the Author(s)

Author specific remarks:

I feel for the authors getting "crushed" between the strongly opposing (even acrimonious) viewpoints of both original reviewers. Although I do not agree with the overall vision of reviewer 1 (e.g. stating "...but the final discussion in this paper is a bit like leaving room for the possibility for intelligent design or for disagreement on global warming." I consider completely overblown) he/she makes some important points. I looked at how the authors dealt with previous criticisms of the two reviewers and think the response is adequate.

I think this revision is acceptable for publication, though I have the feeling that we are in the middle of a muddled discussion and the authors missed a chance of contributing more insight (e.g. what is going on with the two prokaryotic outliers at the top in figures 1C/1D?).

Concerns:

I have some problems with this part of the discussion (though the authors do not have to change anything on my account):

Line 271-276: "The findings eliminate the need to invoke an energetics barrier hypothesis to genome complexity between prokaryotes and eukaryotes; rather, they support the hypothesis of the passive emergence of genome complexity by non-adaptive processes [9,11,12]. As Lynch and Marinov [12] mentioned, the origin of the mitochondrion was not a prerequisite for genome-size expansion, although the origin was a key event in evolutionary history;...." This part and the rest of the paragraph seems to want to convey that cell mass has nothing to do with ATP generation/consumption and that the co-occurrence of the endosymbiont acquisition and the increase of cellular (including genome) complexity is just a coincidence. In what way is it a key event then??? These kind of passages remind me that though Lane and Martin may be wrong about some of the details, overall I still prefer their explanations to the alternatives.

Minor comments (mostly stylistic/language errors).

Line 39: "However, not all prokaryotes have evolved biological complexity." I would say: "However, only some prokaryotes have evolved biological complexity."

Line 281: "The definition of power per genome and power per gene is still controvertible." INCORRECT Language use. "The definition of power per genome and power per gene is still a matter of controversy."

Decision letter (RSOS-191859.R0)

17-Jan-2020

Dear Dr Takemoto

On behalf of the Editor, I am pleased to inform you that your Manuscript RSOS-191859 entitled "Revisiting the hypothesis of an energetic barrier to genome complexity between eukaryotes and prokaryotes" has been accepted for publication in Royal Society Open Science subject to minor revision in accordance with the referee suggestions.

Please find the referees' comments at the end of this email. The Associate Editor and the reviewers have taken a fresh look at your paper and the AE and two of the reviewers are very much in favour of publication. One reviewer is more critical but the AE has considered their views and in particular disagrees with the general criticism that there has not been a sufficient advance. However, both the AE and the reviewers make a number of very useful suggestions for some final improvements of the paper, and we wish you to revise and respond accordingly.

- Ethics statement

- Data accessibility

It is a condition of publication that all supporting data are made available either as supplementary information or preferably in a suitable permanent repository. The data accessibility section should state where the article's supporting data can be accessed. This section should also include details, where possible of where to access other relevant research materials

such as statistical tools, protocols, software etc can be accessed. If the data has been deposited in an external repository this section should list the database, accession number and link to the DOI for all data from the article that has been made publicly available. Data sets that have been deposited in an external repository and have a DOI should also be appropriately cited in the manuscript and included in the reference list.

If you wish to submit your supporting data or code to Dryad (<http://datadryad.org/>), or modify your current submission to dryad, please use the following link:
<http://datadryad.org/submit?journalID=RSOS&manu=RSOS-191859>

- **Competing interests**

- **Authors' contributions**

- **Acknowledgements**

- **Funding statement**

Because the schedule for publication is very tight, it is a condition of publication that you submit the revised version of your manuscript before 26-Jan-2020. Please note that the revision deadline will expire at 00.00am on this date. If you do not think you will be able to meet this date please let me know immediately.

When submitting your revised manuscript, you will be able to respond to the comments made by the referees and upload a file "Response to Referees" in "Section 6 - File Upload". You can use this

to document any changes you make to the original manuscript. In order to expedite the processing of the revised manuscript, please be as specific as possible in your response to the referees.

on behalf of Dr Rees Kassen (Associate Editor) and Steve Brown (Subject Editor)
openscience@royalsociety.org

Associate Editor Comments to Author (Dr Rees Kassen):
Associate Editor

Comments to the Author:

First off, I would like to thank the authors for their patience regarding the evaluation of this manuscript. As explained in my original letter to the authors, the preliminary views by reviewers were so divided that re-review would likely be necessary. This has now happened, with three new reviewers providing comments.

On the basis of these reviews I am recommending the paper be accepted for publication following minor revisions. As you will see two of the reviewers are quite favourable but one is not. The critical reviewer suggests that there is not sufficient advance in this paper over previous work, since the data largely recapitulate previous findings. I disagree. I think the expanded data set, the addition of a phylogenetically-informed analysis, and the revised analyses that include exponential growth rates (rather than just minimal growth rates) provides a more powerful test

of the energetic complexity hypothesis than has previously been available. This is an significant advance, given the limited data that has previously been provided to test the hypothesis.

That said I would encourage the authors to tone down some of their claims about lack of support for the energetic hypothesis and in favour of the non-adaptive hypothesis. The data provided do not, in fact, provide direct evidence for the latter; indeed, this wasn't the point of the paper. As for the argument that there is no evidence for an energetic barrier, well, I would say there is less compelling evidence than has previously been claimed but that the jury is still out on how and why complexity evolved in eukaryotes and not prokaryotes. Reality may be more complicated than either camp would like to think, as hinted at by Reviewer 2.

Specific suggestions for improvement:

1. I would revise this sentence in the abstract: 'Our findings indicate that there is no energetic barrier to genome 25complexity between prokaryotes and eukaryotes, suggesting that the prokaryote-26eukaryote divide is inexplicable from the energetic perspective' to be less strident about the energetic barrier. There could still be a barrier, although it may have been overcome in different ways by prokaryotes and eukaryotes. The difference in complexity is not 'inexplicable', just harder to explain.

2. line 55: the hypothesis is not 'debatable'. It is 'controversial'.

3. Line 274 - see comments on the abstract above. There could still be a barrier, and mitochondria could still play a role in overcoming that barrier, but the results suggest the actual evolutionary pathway to complexity was itself a more complex process than either hypothesis would have it.

4. Finally, Reviewer 3 has some suggestions for improving the text that are valuable as well. Please revise accordingly.

Reviewer comments to Author:

Reviewer: 3

Comments to the Author(s)

Comments on RSOS 191859

Chiyomaru and Takemoto (C&T) revisit the energetics of genome complexity by gathering more data. In the foreground of the file on this paper stands a debate between two L&M teams Lynch&Marionov 2015 (LM15) and Lane&Martin 2010 (LM10). The peer review on the paper has been intense with R1 for the Lynch side talking about "nails in coffins", "nonsensical", a "bizarre rant" versus R2 for the Lane side pointing out in very level headed words that the T&C paper has some issues and that the Lynch side still does not get the point.

The favourable referee R1 weighing in for Lynch says that C&T is "redundant" with earlier work, and that the case for publication is weak ("a difficult call") and that the conclusions are "muddled" because C&T cannot find that L&M10 are really wrong in any way, findings that R1 requests to be restated so as to support a different conclusions. The critical referee R2 has identified fundamental problems with the paper that revisions and new analyses have not resolved.

There has been enough written on this paper, a decision is needed. Referees are supposed to advise Editors. My recommendation to the editor is to decline. The reason is that this referee looked at the RSOS guidelines which state:

Papers should only be recommended for publication if they meet these [five] criteria:

[1] - Original research which has not been published elsewhere.

[2] - Submissions should sufficiently advance scientific knowledge. Negative findings, meta-analyses and studies testing reproducibility of significant work are also encouraged. Repeated experiments will only be considered if they provide a meaningful contribution to the literature. Derivative work will not be considered.

[3] - Conclusions are supported by the data; data supporting the findings of the paper are publicly available.

[4] - Compliance with appropriate ethical standards.

[5] - Experimental protocols/procedure and statistical analysis are performed to a high technical standard and are both methodologically and scientifically sound.

Criterion [1] appears to be met.

Criterion [2], the referees' long comments and the debate between referees in the present file show how incremental the advance is, if there is one. C&T find that L&M10 is reproducible, the first 2 paragraphs of the results state this clearly. Then comes a discussion about cell mass, which is volume because unicellular forms cells have roughly the same density and are 80% water. It is known that some prokaryotes are large, as both L&Ms point out. On lines 200-214 T&C report analyses indicating the eukaryote cells are larger than prokaryote cells (l. 206); that is decidedly not an advance. On lines 215-229 T&C then show that cell mass (volume) also has a phylogenetic component. The result is Table 2 where the reader is challenged to find a strong statistical signal - the values are all very similar.

Criterion [3], the conclusions of L&M10 are supported by the present data, as R1 recurrently points out asking that it not be mentioned, not the conclusions of C&T that L&M15 are supported. The data appears to be available.

Criterion [4], compliance with appropriate ethical standards, there are some accusations about data availability in the text that are hardly acceptable for a scientific paper, in my view. That is not good from an ethics standpoint. The issue on the debate ref 37 vs 38 is 37 having been rejected (37 states so in the text) by the journal that published ref. 11.

Criterion [5], this referee is not an expert in statistics.

The discussion starts out by saying that (l.231) "These results indicate no difference in power per genome and power per gene between prokaryotes and eukaryotes, which is not consistent with Lane and Martin's conclusion that the prokaryotic genome size is constrained by bioenergetics". From there the discussion gets very, very dense, missing the bigger picture presented in L&M10. I think the broad appeal of L&M10 is to the community of physiologists and cell biologists. Eukaryotes respire in mitochondria, prokaryotes respire at the plasma membrane, eukaryotes are complex, prokaryotes are not, eukaryotes have meiosis, prokaryotes do not.

On the bottom line, as this reader/referee sees it, C&T say that mitochondria had no role in evolution. L&M10 are saying that symbiosis might be an important force in evolution that can create novelty, while L&M15 are saying that that all evolution needs in order to create novelty is variation in population size. C&T report data that R1 finds redundant and R2 finds to be entirely missing the point.

I think that R2 has the better arguments, hence my recommendation above. I really like symbiosis in evolution, hence I like L&M10. Martin published a good paper entitled "Symbiogenesis, gradualism and mitochondrial energy in eukaryote evolution" on this debate in an obscure but readily accessible open access journal. He pointed out that mitochondria produce about 95% of the ATP in a respiring cell, but comprise only 10% of the cell's volume, releasing constraints on the remaining 90% of the cell (mostly cytosol) to evolve other functions, structural proteins rather than enzymes, while leaving everything under that control of a handful of more genes whose expression not copy number is increased. How can that be irrelevant for evolution? Discussing that would help to explain what the actual issue is here, because it was the role of mitochondria in the origin of complexity of eukaryotic cells that was at the heart of the L&M10 paper.

Especially the Discussion is not focussed on science, it is about who did what to whom, which is immaterial for understanding the role of mitochondria in eukaryote origin. The "debate" as presented in the C&T paper is hardly enlightening, as both R1 and R2 conclude, albeit for very different reasons.

Reviewer: 4

Comments to the Author(s)

I've checked all the boxes because they are required, but I really have not had the time to reason my way through the arguments here, in which I am not in any case an expert. My concerns about Lane and Martin 2010 are nevertheless emphasized by authors' Fig. 1, which shows overlap between prokaryotes and eukaryotes, however measured. This seems to me to make the only point that needs to be made: we might easily imagine a large-genomed, high-energy prokaryote "evolving into" a small-genomed, low-energy eukaryote. Indeed that's what most sensible folks would imagine - not that something like the average modern prokaryote "evolved into" something like the average modern eukaryote. Thus comparisons of averages seem to me beside the point. Moreover, there are eukaryotes without mitochondria that have most of the bells and whistles of other eukaryotes without the mitochondrion-derived energy. So such a scenario is possible.

Reviewer: 5

Comments to the Author(s)

Author specific remarks:

I feel for the authors getting "crushed" between the strongly opposing (even acrimonious) viewpoints of both original reviewers. Although I do not agree with the overall vision of reviewer 1 (e.g. stating "...but the final discussion in this paper is a bit like leaving room for the possibility for intelligent design or for disagreement on global warming." I consider completely overblown) he/she makes some important points. I looked at how the authors dealt with previous criticisms of the two reviewers and think the response is adequate.

I think this revision is acceptable for publication, though I have the feeling that we are in the middle of a muddled discussion and the authors missed a chance of contributing more insight (e.g. what is going on with the two prokaryotic outliers at the top in figures 1C/1D?).

Concerns:

I have some problems with this part of the discussion (though the authors do not have to change anything on my account):

Line 271-276: "The findings eliminate the need to invoke an energetics barrier hypothesis to genome complexity between prokaryotes and eukaryotes; rather, they support the hypothesis of the passive emergence of genome complexity by non-adaptive processes [9,11,12]. As Lynch and Marinov [12] mentioned, the origin of the mitochondrion was not a prerequisite for genome-size expansion, although the origin was a key event in evolutionary history;...." This part and the rest of the paragraph seems to want to convey that cell mass has nothing to do with ATP generation/consumption and that the co-occurrence of the endosymbiont acquisition and the increase of cellular (including genome) complexity is just a coincidence. In what way is it a key event then??? These kind of passages remind me that though Lane and Martin may be wrong about some of the details, overall I still prefer their explanations to the alternatives.

Minor comments (mostly stylistic/language errors).

Line 39: "However, not all prokaryotes have evolved biological complexity." I would say: "However, only some prokaryotes have evolved biological complexity."

Line 281: "The definition of power per genome and power per gene is still controvertible." INCORRECT Language use. "The definition of power per genome and power per gene is still a matter of controversy."

Author's Response to Decision Letter for (RSOS-191859.R0)

See Appendix B.

Decision letter (RSOS-191859.R1)

21-Jan-2020

Dear Dr Takemoto,

It is a pleasure to accept your manuscript entitled "Revisiting the hypothesis of an energetic barrier to genome complexity between eukaryotes and prokaryotes" in its current form for publication in Royal Society Open Science. The comments of the reviewer(s) who reviewed your manuscript are included at the foot of this letter.

on behalf of Dr Rees Kassen (Associate Editor) and Steve Brown (Subject Editor)
openscience@royalsociety.org

Appendix A

Response to Editor (Prof Dr Rees Kassen)

Editor's Comment:

Your paper has now been seen by two reviewers, one of whom (Nick Lane) has waived their right to anonymity, as you will see. On the basis of these reviews, I am recommending the paper be rejected and resubmission be allowed and, indeed, encouraged.

The reviewers present strongly contrasting views of the manuscript. This is perhaps not surprising given the controversy - and at times vitriol - surrounding the hypothesis you are testing. In crafting my recommendation, I sought to take these highly polarized views into account.

The key observation demanding explanation is that eukaryotes are so much more complex, in terms of genome size and architecture as well as other features (cell morphology, multicellularity, etc...), than prokaryotes. The controversy in the literature is over why. Lane and Martin (2010) argued that complexity evolved as a consequence of the endosymbiosis that gave us mitochondria, which increased the supply of energy available to the cell and so permitted complexity to evolve. Lynch and Marinov (2015) argued, in response, that complexity is a consequence of weak selection against excess DNA driven by an increase in eukaryotic cell size, and so decrease in effective population size. The current paper seeks to confront the Lane and Martin hypothesis directly with more data and a more sophisticated analysis. It does not take on directly the Lynch and Marinov hypothesis.

Our Response:

We sincerely thank you for handling the reviewing of our manuscript. According to your and reviewers' comments and suggestions, we largely revised the manuscript. Thanks to your kind advices and suggestions, the quality of this study largely increased. We believe that the revised manuscript meets the criteria for publication. We would appreciate if you handle the reviewing of our manuscript again.

Editor's Comment:

The results are interesting - the apparent increase in energy supply posited by Lane and Martin appears to be much smaller once cell size and phylogeny are taken into account, weakening the energy-supply hypothesis. Nick Lane has, correctly I think, pointed out that at least two features of the current data set must be incorporated to provide a more compelling test: data on ploidy levels and some measure of exponential growth rate (rather than minimum metabolic rate alone). I concur, if only because doing so would provide a more direct comparison with the original work. A revised ms should include responses to all other criticisms as well, of course.

I therefore invite the authors to perform these analyses and resubmit their results. At that time I would seek to identify at least one, if not two, additional reviewers who are less intellectually (and perhaps emotionally) invested in the results.

Our Response:

We are really happy to receive the positive comment from you. Our dataset has already included data on ploidy levels and metabolic rates measured at the exponential or logarithmic growth phase, although we used the minimum metabolic rates in the previous version of our manuscript. We further added the comments on metabolic rate measurement (growth phase, culture age, etc.; see electronic supplementary material, dataset S1). We used these data for a more direct comparison with the original work and obtained the similar result (i.e., limited contribution of power per genome and power per gene to the prokaryote-eukaryote dichotomy). We collected data on ploidy in organisms as much as possible; however, we assumed that prokaryotes whose ploidy level had not been reported in any previous studies were monoploid. However, this limitation may pose little problem because similar tendencies (i.e., limited contributions of power per genome and power per gene to the prokaryote–eukaryote dichotomy) were observed in standard regression analyses even if we removed prokaryotes whose ploidy level had not been reported (electronic supplementary material, table S2). Our revised manuscript also includes our responses to all other criticisms (see below for details).

=====
Response to Reviewer 1
=====

Reviewer's Comment:

This paper reports on a set of analyses that debunk the claim of Lane and Martin that the origin of the mitochondrion provided a massive boost to the bioenergetic capacity of eukaryotes, spawning in its wake evolutionary diversification of this group. There are two ways of looking at the paper. First, it may be argued that Lynch and Marinov (in two papers) have already provided substantial evidence against the Lane-Martin hypothesis. Second, given Lane and Martin's (especially the latter's) somewhat vitriolic defense of the undefensible, it may be argued that a final nail in the coffin is in order. The authors do take a somewhat different tack than Lynch and Marinov, so this may be viewed as useful. I am not sure the approaches get at the problem as directly as was done by Lynch and Marinov, and in places I think the authors are far too deferential to Lane and Marinov. All in all, this is a difficult call, and I would like to see some modifications, but I don't see anything fundamentally wrong with the paper.

Our Response:

We really appreciate your kind review. According to your comments, the manuscript was largely revised. Thanks to your kind advices and suggestions, the quality of this study largely increased. We believe that the revised manuscript meets the criteria for publication. We would appreciate if you review our study again.

Reviewer's Comment:

1. There are a number of places in which the terminology used by the authors is unclear, e.g., the "minimum mass-specific metabolic rate to maintain orderliness of unit live mass". Indeed, in general, I think the authors need to address more directly their whole reliance on metabolic-rate data. What does metabolic rate really mean in evolutionary terms? Ultimately, one would like to know how much different organisms actually put into biomass, and metabolic rate doesn't tell us anything about that – it is simply some measure of how much energy is burned, without providing any direct insights into the benefits. Admittedly, Lane and Martin focused on this kind

of currency, and so the authors might be forgiven for going down the same route, but I think it would be best if they addressed the issue of the utility of this measure head on. This is the whole reason that Lynch and Marinov elected to work on cell growth (while not ignoring basal metabolic costs).

Our Response:

According to comments of Editor and Reviewer 2 (Dr Nick Lane), we instead used the maximum mass-specific metabolic rate for a more direct comparison with the original work; thus, we removed the description of "minimum mass-specific metabolic rate". This study does not aim to show the validity of the usage of metabolic rate in evolutionary studies. As you pointed out, rather, we just considered metabolic rate for a comparison with the original work. We strongly recognize that the usage of metabolic rate as a measure of power production is uninformative from an evolutionary perspective. We emphasized such a limitation in Introduction and Discussion sections; moreover, we highlighted the need for more suitable measures of power production (e.g., the number of ATP=>ADP turnovers).

Reviewer's Comment:

2. The authors do quite a bit of waffling about ploidy levels, but I do not really see the huge issue here – after all the cost at the DNA level is quite small to everything else (RNA and protein, as pointed out by Lynch and Marinov), and the issues in prokaryotes are quite different than those in eukaryotes, as in the former ploidy level is transient, and in the latter, there is a wide range of variation in the meaning of the term – from simultaneously segregating multivalents to ancient polyploidization events. To me, this is all a bit of a distraction from the main point, and not likely to be of major relevance.

For similar reasons, I find the use of the currency of “power per gene” as bordering on nonsensical, as it has no obvious relationship to fitness, although I appreciate that the authors are somewhat stuck with this term in their analysis, which attempts to validate the Lane-Martin results. I do think it would still be useful to have a short commentary on the meaningfulness of such an index. Again, I understand that the authors are attempting to replicate the Lane-Martin results, and showing this cannot be done, but I think they are missing the opportunity to weigh in a bit more deeply on the general issues.

Our Response:

We completely agree with you; however, we carefully considered ploidy levels as this study aims to revisit the Lane-Martin hypothesis. As you mentioned, moreover, it is hard to say that the currency of power per gene is nonsensical from our result. Instead, we clearly explained the limitation of metabolic rate (i.e., why the usage of metabolic rate is not helpful as a measure of power production) in Discussion section; moreover, we emphasized the need for more suitable measures of power production (e.g., the number of ATP=>ADP turnovers).

Reviewer's Comment:

3. I find it quite surprising that the authors were unable to get the original data sets that Lane and Martin used in their Nature paper, which seems to be bordering on the unethical from the standpoint of journal requirements. What is particularly concerning here is that Martin has recently attacked Lynch and Marinov in a Biology Direct paper in a somewhat bizarre rant about

data unavailability when the data were in fact available. At the very least, the authors here should reference the response by Lynch and Marinov (in Biology Direct as well, which clarifies the situation), and not simply cite reference 17, which will lead the uninformed reader to think there is a legitimate controversy here.

Our Response:

We completely agree with you. As you can see from the review report of Dr Nick Lane, he has lost the original data. More careful examinations would be possible if he regenerate the data. According to your comment, we cited Lynch, M. & Marinov, G. K. 2018 Response to Martin and colleagues: Mitochondria do not boost the bioenergetic capacity of eukaryotic cells. Biol. Direct 13, 9–10. (doi:10.1186/s13062-018-0228-3) and clearly mentioned that there is a legitimate controversy here. We also emphasized the importance of open data here.

Reviewer's Comment:

4. In the discussion, the authors state that their results support Lane-Martin's contention that genome size is constrained by bioenergetics. This is a pretty surprising statement. Perhaps the authors are attempting to be nonconfrontational, but I don't see this at all. There is nothing at all in the current paper that suggests that bioenergetics is of any relevance to the prokaryotic-eukaryotic divide, and repeating this statement is not helpful. It seems that this statement derives from a "power per gene" argument, which is in turn debunked by the authors' own size-scaling analyses. In the end, the authors even state that they don't see that their analyses "contradict the traditional hypothesis". It is unclear what traditional hypothesis is being referred to, but if it is the Lane-Martin hypothesis, this makes little sense, as this hypothesis has hardly been embraced, and a statement like this, which is completely contrary to the paper, will only cause the L-M myth to live on.

In short, the authors present some quite clear-cut results, somewhat redundant to Lynch and Marinov, but done in a different way, but things become muddled at the end when they seem to somehow hold on to the rope for some credence in the Lane-Martin hypothesis even after removing all such credence...

Our Response:

Sorry for misleading. We wanted to say as follows: these results finally indicate no difference in power per genome and power per gene between prokaryotes and eukaryotes, inconsistent with Lane and Martin's conclusion that prokaryotic genome size is constrained by bioenergetics. The simple comparison tests (figures 1a and 1b) indicated that power per genome and median power per gene of eukaryotes were greater than those of prokaryotes; however, the observed differences were artifacts due to no consideration of the effects of cell mass and phylogeny. We largely revised Discussion section to emphasize our main finding (i.e., limited contribution of power per genome and power per gene to the prokaryote-eukaryote dichotomy).

Reviewer's Comment:

Although I am in favor of publishing this paper, I do think these loose ends need substantial tightening up. The authors have some very clear conclusions, so why leave the door open to debate, especially given the prior evidence against Lane-Martin. I acknowledge that science

profits from debate, but the final discussion in this paper is a bit like leaving room for the possibility for intelligent design or for disagreement on global warming.

Our Response:

Thank you for your suggestion. We largely revised the manuscript to emphasize our main finding (i.e., limited contribution of power per genome and power per gene to the prokaryote–eukaryote dichotomy). However, our findings cannot ****entirely**** discount the traditional hypothesis because of the limitations in our data analyses (e.g., extremely polyploid giant bacteria were not considered in this study because of no data although our dataset has prokaryotes with ploidy level of at most 218); thus, we wrote the manuscript with a weak expression. Rather, we emphasized the importance of rigorous evidential and statistical support for debate in the prokaryote–eukaryote divide.

Response to Reviewer 2 (Prof Dr Nick Lane)

Reviewer's Comment:

I think it is necessary that I identify myself at the outset on this review: I am Nick Lane, one of the authors of the original Lane & Martin Nature 2010 paper that is under reanalysis here. This means that I have a bias, which it is only fair to recognise. For that reason I do not think that my comments should carry as much weight as the other referees for this paper. However, I should also say upfront that I do not agree with the conclusions of the paper as written. My comments here are sincerely meant to improve the paper. I would hope that they would at least modify the tone and some of the content of a revised MS.

Our Response:

We really appreciate your kind review. It is our honor to receive many comments and suggestions from you, a prominent researcher of evolutionary biology. According to your comment, we largely revised the manuscript. Your comments and suggestions largely enhanced the quality of our manuscript. Please remember that it is not as if we want to discount your hypothesis. Rather, we just would like to emphasize the importance of rigorous evidential and statistical support for debate in the prokaryote–eukaryote divide. We strongly recognize that your Nature paper is positioned as a proposal of a hypothesis rather than data analysis.

Reviewer's Comment:

I accepted reviewing this paper because it is obviously important to me. This is a heated field and I dislike the tone of the debate over the last few years, especially between Bill Martin and Mike Lynch, who stands accused of fabricating data. That is as may be. I have other problems with the position of Lynch. The present paper seems to support the views of Lynch and others. If Bill Martin and I were wrong, I would prefer to be the first person to accept that publicly, hence my agreement to review the paper. Science should aim for the truth, not winning arguments. So I have reviewed this paper with the truth in mind and the intention to be as honest as possible: should I in all honesty put up my hand and say 'we were wrong, our position is now untenable?' The answer on the strength of this analysis is 'no'. I still think our hypothesis is correct, and to

my mind this paper misses the point. That is a shame, as the intention of analysing a wider range of data with improved statistical methods is good and potentially important. But the parameters used matter a great deal. Unfortunately the authors have ignored what are in my view the most important parameters.

Our Response:

In short, to revisit your hypothesis, the maximum mass-specific metabolic rates were used to estimate energy supply (electronic supplementary material, dataset S1); specifically, they mainly correspond to mass-specific metabolic rates measured at the exponential or logarithmic growth phase and summit metabolic rates. Moreover, we also considered ploidy level (highest bacterial ploidy level is 218). We obtained similar conclusion (i.e., limited contribution of power per genome and power per gene to the prokaryote–eukaryote dichotomy). We collected data on ploidy in organisms as much as possible; however, we assumed that prokaryotes whose ploidy level had not been reported in any previous studies were monoploid. However, we believe that this limitation may pose little problem because similar tendencies (i.e., limited contributions of power per genome and power per gene to the prokaryote–eukaryote dichotomy) were observed in standard regression analyses even if we removed prokaryotes whose ploidy level had not been reported (electronic supplementary material, table S2). As you pointed out, however, our findings may not entirely discount the traditional hypothesis because of the limitations of data analyses (see the following responses to your specific comments for details); rather, they emphasize the need for more careful examinations.

Reviewer's Comment:

It seems to me that there has been a systematic error in interpreting what was meant by 'energy per gene' in the original Nature paper, and this error is perpetuated here. It might be that we were not sufficiently clear in that paper, and indeed I have published several other papers that aimed to make at least my own position more clear. However they are rarely cited in comparison. Two of those papers are cited here, but their arguments are not touched upon at all. For example, there are three section headings in the relevant part of Lane 2014 (Cold Spr Harb Persp Biol), each of which is supported by brief argument. They are as follows: 'Scaling of cell volume and genome copy number'; 'The requirement for core bioenergetic genomes'; and 'Endosymbiosis in prokaryotes gives rise to extreme genomic asymmetry'. None of these tenets are even mentioned in the present paper, or previously by Lynch.

Our Response:

We clearly mentioned that you stated that power per gene means the cost of expressing the gene or energy availability (i.e., supply, not demand) per gene and explained why the discrepancy (the debate on PNAS) occurred. Moreover, we emphasized the importance of extreme polyploidy (see the following responses to your specific comments for details). We also discussed the relationship between cell volume and genome copy number. In particular, we found that the prokaryotic ploidy level was correlated with cell mass (Spearman's rank correlation coefficient $r_s = 0.43$ and the associated p-value $p = 0.0092$) in our dataset.

Reviewer's Comment:

The key point for me is this: the original Lane & Martin Nature paper was not primarily an analysis (as described here) but a hypothesis, and published as such. That hypothesis postulates

that genes are needed to control respiration, and that only an endosymbiosis can put the right genes in the right place, right next to bioenergetic membranes. The core of the argument is that mitochondria always have a genome, which has been whittled away to nearly nothing, but never lost completely except when ox phos was also lost. We follow John Allen in arguing that these genes are needed to control respiration. That in turn predicts that giant bacteria such as *Epulopiscium* and *Thiomargarita* should have lots of copies of their genome, and indeed they are found to exhibit extreme polyploidy, with tens or hundreds of thousands of copies of their complete genome. When the costs of expressing all that genomic weight is taken into consideration (specifically the costs of gene expression, meaning protein synthesis), then giant bacteria have very much lower energy availability per haploid gene or genome than eukaryotes. Energy per gene always meant the power for gene expression (not gene number, as incorrectly asserted by Lynch).

Our Response:

Yes, we strongly recognize that your Nature paper is positioned as a proposal of a hypothesis rather than data analysis. We globally revised the manuscript to emphasize that. The title was also changed. We have considered ploidy level in our data analyses (see the following responses to your specific comments for details). Moreover, we may need to consider extreme polyploid bacteria such as *Thiomargarita* and *Epulopiscium* with lots of copies of their genome [6,8,34,39] although our dataset included prokaryotes with relatively high ploidy level (highest value = 218). However, we needed to exclude these species in our data analyses because the parameters required in data analyses were unavailable and/or ambiguous. For example, phylogenetic regression analyses requires accurately annotated genomes. Of course, this a limitation in our study. We clearly mentioned and discussed this limitation. Moreover, we emphasized that our findings do not ****entirely**** discount the your hypothesis.

Just to tell you, *Thiomargarita* power per cell in your Nature paper is wrong; however, this may not affect your conclusion. Following Schulz and de Beer paper, you used, *Thiomargarita* metabolic rate is about 12 [pmol / cell / h]; thus, *Thiomargarita* power per cell is 1700 [pW] (using 20 J/mLO₂ and 25.5 L/mol; ideal gas molar volume at 25°C and 1 bar), not 500 [pW] (Schulz also confirmed this; personal communication).

Reviewer's Comment:

The present paper does not mention mitochondrial DNA or extreme polyploidy even once (and in all his papers, Lynch has not mentioned mtDNA or extreme polyploidy once). We have said nothing about the efficiency of respiration (as implied here) - we have talked about the ability of cells to SUPPLY enough energy to meet the costs of gene expression when scaling up to eukaryotic volumes. We do not dispute that the energy required for gene expression scales allometrically (or isometrically; it matters little) with the area of bioenergetic membranes. Plainly there is a relationship between the area of bioenergetic membranes and the metabolic rate, and between the number of ribosomes and cell volume and gene expression, as argued by Lynch. There is no reason to think that it should cost more or less to express a gene in bacteria or eukaryotes. Certainly there is a continuum, but there is little if any overlap on this continuum. What does change is supply: the ability of cells to produce enough energy to scale up indefinitely. We contend that to scale up to eukaryotic size requires extreme polyploidy (as with mtDNA in eukaryotes). It is precisely this argument that has been ignored here and in other

recent critical literature, despite being explicitly laid out in a reply to Lynch and Marinov (Lane & Martin PNAS 2016), not cited here, and at greater length in Lane J Theoret Biol 2017 (also not cited here). The further point that eukaryotes have a genomic asymmetry, in which tiny mitochondrial genomes support energetically a massively expanded nuclear genome is missing completely.

Our Response:

We clearly mentioned that you stated that power per gene means energy availability per gene: supply, not demand. We considered ploidy levels and obtained similar conclusion (see the following responses to your specific comments for details). Our dataset included prokaryotes with relatively high ploidy levels (218, the highest value). However, We cited these papers not cited in the previous version of our manuscript and emphasized the need for extreme polyploid bacteria such as *Thiomargarita* and *Epulopiscium* with lots of copies of their genome in Discussion section.

Reviewer's Comment:

Below I give some specific comments on the MS.

Our Response:

Thanks. We provide a point-by-point description of the changes in response to your comments and suggestions.

Reviewer's Comment:

Abstract: 'the hypothesis is based on a biased evaluation'. Certainly it is based on a small sample size but the hypothesis is based on the reasoning outlined above, and the sample was a small part of that. A large part included reasoning about surface-area-to-volume scaling to control for volume and the costs of extreme polyploidy in giant bacteria. One of the reasons that we had a smaller sample size was precisely that we needed to know the ploidy as well as genome size and cell volume. This has been completely overlooked here.

Our Response:

We removed this misleading. In particular, we emphasized that the evaluation based on relatively small dataset was due to data availability at the time.

Reviewer's Comment:

Abstract: 'Bioenergetically efficient'. We do not claim this (see above). I imagine that prokaryotes are more bioenergetically efficient than eukaryotes. Our claim is that we can explain the absence of what is in fact never seen: we do not see large, eukaryotic-sized prokaryotes with eukaryotic-sized genomes and orders of magnitude more ribosomes etc, combined with internal endomembrane systems, dynamic cytoskeletons, etc. What we see instead is extreme polyploidy with metabolic inertia in the internal volume. None of this is discussed here.

Our Response:

We simply mentioned that our findings indicate no energetic barrier to genome complexity between prokaryotes and eukaryotes. Note that our study is limited to power per genome and

power per gene based on metabolic rate. We discussed the later part in Discussion section because of lack adequate space for this discussion.

Reviewer's Comment:

Introduction: 'Lane & Martin used data on genome size and metabolic rate...' this omits to say that we also used ploidy, which is the critical factor as discussed above. So this presents a very partial view already, and I reiterate that this one only part of a hypothesis that was largely reasoned rather than based on data analysis.

Our Response:

We also clearly mentioned ploidy level here. Moreover, we emphasized that your Nature paper is positioned as a proposal of a hypothesis rather than data analysis.

Reviewer's Comment:

'Lynch has pointed out that the increase in genome complexity can be explained by non-adaptive evolutionary processes'. This is neither referenced nor frankly subjected to an iota of thought. Really? So the beauty of the eukaryotic nucleus, with the nuclear pore complexes, elastic lamina, nucleolus, chromosomal packing, meiosis, mitosis, splicing etc etc (I could go on and on) is a result of drift in small populations. It seems to me that is a far weaker claim and should be subject to some serious analysis. Instead it is trotted out without question. But then two sentences later I read that 'the Lane-Martin hypothesis has severe limitations' and was 'based on a biased evaluation of a limited number of species'. See all of the above. I am beginning to wonder if the authors are in some way associated with Lynch because this argument is extremely biased.

Our Response:

Please note that this is limited to genome complexity (genome size and gene number). Lynch clearly mention the passive emergence of genome complexity by nonadaptive processes in [Lynch, M. 2007 The frailty of adaptive hypotheses for the origins of organismal complexity. Proc. Natl. Acad. Sci. 104, 8597–8604]. Lynch also mention that a plausible scenario is that the full eukaryotic cell plan emerged at least in part by initially nonadaptive processes made possible by a very strong and prolonged population bottleneck in [Lynch, M. & Marinov, G. K. 2017 Membranes, energetics, and evolution across the prokaryote-eukaryote divide. Elife 6, 1–30]. This hypothesis is also still debatable, but such a statement exists at least. In addition, we are not associated with Lynch.

Reviewer's Comment:

'Eukaryotes are no more bioenergetically efficient than eukaryotes'. Again, see above. Lynch missed the point and so do the authors here.

Our Response:

We simply mentioned that the authors found no energetic difference between eukaryotes and prokaryotes.

Reviewer's Comment:

Aims of paper - fails to mention ploidy at all and so completely misses the point of the paper they say they are reanalysing.

Our Response:

We clearly mentioned ploidy here.

Reviewer's Comment:

Materials and Methods. 'Following a previous study...' again a failure to mention ploidy, which was the key message of the previous study.

Our Response:

We have mentioned ploidy in Sec 2.2.

Reviewer's Comment:

'We used minimum mass-specific metabolic rate'. To my mind this is a bizarre decision and undermines the meaning of the whole study. The idea is to test whether there are energetic limitations, and yet the authors use minimum metabolic rate? Limitation means pushing to the limits. Minimum metabolic rate pushes nothing to the limits except possibly how low metabolic rate can get before an organism enters dormancy. It is amazing to me, given this decision, that the authors find any difference at all between eukaryotes and prokaryotes. Dormancy is close to zero metabolic rate regardless of phylogeny. So this study necessarily compresses any differences discussed by Lane & Martin, but this important difference is not highlighted or discussed properly anywhere in the paper - the conclusion that the differences are less extreme than previously reported is presented without this major proviso. Drawing from the same dataset, Lane & Martin had used metabolic rate during the exponential growth phase, which gives some sense of the normal energetic limits of growth. I have to ask myself: is this paper making criticisms of Lane & Martin that I should be worrying about, and the answer at this point is 'no'. This choice of minimum metabolic rates in my view renders the rest of the arguments pointless.

Our Response:

We really thank this comment. We used the maximum mass-specific metabolic rate, which mainly correspond to a mass-specific metabolic rate measured at the exponential or logarithmic growth phase, in order to estimate metabolic supply (electronic supplementary material, dataset S1 and tables S1) and obtained similar results. In particular, we found no difference in power per genome and power per gene between eukaryotes and prokaryotes when statistically controlling for confounding effects of cell size and phylogenetic signals. However, your comment "Lane & Martin had used metabolic rate during the exponential growth phase" may be an overstatement. The dataset you used just include metabolic rates of prokaryotes growing in various substrates in [Makarieva, A. M., Gorshkov, V. G. & Li, B.-L. 2005 Energetics of the smallest: do bacteria breathe at the same rate as whales? Proc. R. Soc. B Biol. Sci. 272, 2219–2224]); thus, they may not include metabolic rates mastered at the exponential growth phase because there are several growth phases.

Reviewer's Comment:

'For prokaryotes, ploidy levels were generally set to one because bacteria are generally assumed to be monoploid during very slow growth.' Again: are the authors doing a reanalysis or are they deliberately setting out to disprove a hypothesis on what I would see as utterly spurious grounds (again, hidden in the small print of the methods section). 'Some bacteria may be oligoploid or

polyploid' hardly gives any indication that the giant bacteria *Epulopiscium* and *Thiomargarita*, which formed the basis of the argument in Lane & Martin, exhibit 'extreme polyploidy' (not our term). These bacteria can have as many as 200,000 copies of their complete genome. But both of these giant bacteria are excluded from the present analysis with no reason given at all - they are simply not mentioned. The concept 'energy per haploid gene' or 'genome' means for each identical copy, so the metabolic rate is divided by the copy number as well as genome size. This is by far the greatest source of disparity between the papers, but is nowhere is this difference even mentioned. At this stage I am wondering if the authors have a deliberate intention to obscure the truth. The highest bacterial ploidy I could discover in the SI was about 20 (I no longer have access to the SI).

Our Response:

Sorry for misleading. We revised the descriptions. In short, we have used ploidy level when calculating power per gene and power per genome. For prokaryotes, we collected ploidy levels from the literature (electronic supplementary material, dataset S1). The highest bacterial ploidy level is 218 (*Synechocystis*) in our dataset. However, we assumed that prokaryotes whose ploidy level has ****not**** been reported in any previous studies were monoploid. Of course, this assumption is a limitation. However, we believe that this limitation may pose little problem because similar conclusion was obtained even if we remove such prokaryotes (i.e, prokaryotes whose ploidy level had not been reported in any previous studies). We excluded giant bacteria because sources are not reliable for us (for example, *Thiomargarita* genome is not completed; moreover, *Thiomargarita* ploidy level is from personal communication). Moreover, phylogenetic regression analyses require highly annotated genomes. Of course, the fact that giant bacteria cannot be considered in data analyses is a limitation. We clearly mentioned and discussed this limitation in Discussion section.

Reviewer's Comment:

Results. Given that the authors use minimal metabolic rates and assume of general ploidy of 1, I am surprised that they still find a difference in energy per gene of around 350-fold.

Our Response:

We reanalyzed this using the maximum metabolic rates.

Reviewer's Comment:

Correcting for cell mass. We clearly stated in the original paper and in several other papers that the difference between bacteria and eukaryotes primarily reflected the difference in cell volume. We then corrected for cell volume theoretically on the basis of scaling arguments. That exercise was (we stated clearly) simplistic and assumed respiration across the surface of a sphere, but the argument pointed to the requirement for extreme ploidy in scaling up volume over 15,000-fold (the difference in mean volume between the bacteria and the archaea in our study. Our argument was that the larger the bacterium the greater the ploidy ought to be. I stated it even more clearly in Lane CSHP Biol 2014: 'The scaling of polyploid genomes with prokaryotic cell volume is ultimately what prevents both bacteria and archaea from attaining eukaryotic complexity (but has been ignored by some; Lynch and Marinov, 2015, 2017). Yet the logistic regression analyses performed here to control for the effects of cell mass did not include ploidy - the one factor that

we explicitly predicted should correlate with cell mass. While it is true that there is not much reliable data on ploidy in bacteria, the authors have not even discussed the issue.

Our Response:

Sorry for misleading. We revised the descriptions. In short, we also have used ploidy level in regression analyses (see Sec. 2.3). For prokaryotes, we collected ploidy levels from the literature (electronic supplementary material, dataset S1). However, we assumed that prokaryotes whose ploidy level has ****not**** been reported in any previous studies were monoploid. Of course, this assumption is a limitation. We clearly discussed this limitation in Discussion section. However, we believe that this limitation may pose little problem because similar results were obtained even if we remove such prokaryotes (i.e, prokaryotes whose ploidy level had not been reported in any previous studies).

Reviewer's Comment:

Discussion. 'We used the minimum mass-specific metabolic rates to accurately estimate cellular maintenance costs'. I reiterate my point from above: cellular maintenance costs have nothing whatsoever to do with the question at hand.

Our Response:

We removed the description, as we used the maximum metabolic rates.

Reviewer's Comment:

'We were informed the dataset was presently unavailable'. I had responded to the email from Kazuhiro Takemoto to say: 'Thanks for your interest. I'll try to dig it out - I don't recall a dataset as such for that table, more simply literature sources for genome size, ploidy, metabolic rates etc. I'll need to relocate them as it was quite a long time ago now. And I'm mostly travelling in the next 2-3 weeks but I'll do my best to get it to you ASAP.' I regret that I failed to get back to Kazuhiro after my travels, but had been hard pressed for a long period. I would have had to mine my original sources in much the same way that they would: all our data were sourced from the literature and referenced appropriately. I do not think that what the text says about this is strictly accurate: it implies that we were deliberately concealing data. Nor should I be acknowledged for 'useful comments on data availability'. That implies I am complicit somehow in this paper, which I am not.

Our Response:

Sorry. We did not mean to say that you were deliberately concealing data. We revised this part. We emphasized your help. We removed your name from Acknowledgements section. However, we again request the original dataset for ****open science****. In particular, we want to know the ploidy levels of 55 prokaryotes, you mention in Nature paper. We are wondering why you had been able to find the ploidy levels of so many prokaryotes. We also searched for literature as much as possible; however, we could not find the ploidy levels of 55 prokaryotes. We expected that you attached the original dataset of your Nature paper to this review report; however, we could not find the dataset.

Reviewer's Comment:

'Lack of difference in power per genome and power per gene between prokaryotes and eukaryotes of similar mass.' The Lane-Martin prediction is that there should be no difference between small eukaryotes and prokaryotes (where ploidy is 1), but the larger the bacteria, the greater the ploidy and the greater the difference should be - so long as ploidy is taken into consideration, and so long as metabolic rate during exponential growth rather than minimum maintenance are taken into consideration. See above.

Our Response:

We used the maximum metabolic rate. Ploidy level was also considered. We obtained similar conclusion (i.e., no difference between eukaryotes and prokaryotes when controlling for the effects of cell mass and phylogeny). In this study, we assumed that prokaryotes whose ploidy level has ****not**** been reported in any previous studies were monoploid. However, we believe that this limitation may pose little problem because similar results were obtained even if we remove such prokaryotes (i.e, prokaryotes whose ploidy level had not been reported in any previous studies).

Reviewer's Comment:

'Our findings are inconsistent with the idea that cells with greater internal complexity impose greater energy demands'. This is absurd. Of course large complex cells have greater energy demands. The argument here should be that large complex bacteria would also have greater energy demands. But the problem is that there are NO large, complex bacteria, so this contention cannot be tested through the analysis of any databases. This lack of overlap is unclear in the present paper (but is clear in Lynch papers) because minimum maintenance metabolic rates were used here, without considering ploidy. These difference would surely have been far more obvious had the analysis used exponential growth rates and extreme polyploidy.

Our Response:

As mentioned above, we used the maximum metabolic rate. Ploidy level was also considered. We obtained similar conclusion (i.e., no difference between eukaryotes and prokaryotes when controlling for the effects of cell mass and phylogeny). In this study, we assumed that prokaryotes whose ploidy level has ****not**** been reported in any previous studies were monoploid. However, we believe that this limitation may pose little problem because similar results were obtained even if we remove such prokaryotes (i.e, prokaryotes whose ploidy level had not been reported in any previous studies).

Reviewer's Comment:

'Eukaryotes are not bioenergetically more efficient than prokaryotes'. We never said they were. See above.

Our Response:

Revised. See above.

Reviewer's Comment:

The whole section on the similarity of mean MINIMUM mass-specific metabolic rates is not relevant to the question in my view as it minimises the differences between cells as well as ploidy levels.

Our Response:

We removed this section.

Reviewer's Comment:

'We only considered organisms for which complete genome sequences were available.' Another strange choice. The only reason to use a complete genome sequence is to confirm the number of open reading frames. But if an incompletely sequenced genome is thought to have 4000 open reading frames, it is unlikely that the margin for error would be more than say 20% - perhaps there might be 5000 or 3000 open reading frames, but the margin of error is small. But if the actual ploidy is 2, then the estimate of energy per genome is out by 2-fold. In the case of giant bacteria, it would be out by many thousands-fold. So the use of full genome sequence data is irrelevant, and in fact excludes giant bacteria that have not been fully sequenced yet, but whose ploidy has previously been robustly estimated.

Our Response:

Accurate genome size and gene number are required to accurately calculate power per genome and power per gene, defined in your Nature paper. Thus, we considered organisms for which complete genome sequences were available. Moreover, phylogenetic regression analyses require highly annotated genomes. Thus, we do not think this is strange in scientific context. As you pointed out, however, our study has a limitation that giant bacteria cannot be considered in data analyses. We clearly mentioned and discussed this limitation in Discussion section.

Reviewer's Comment:

The only comment on ploidy in the discussion is the following: 'We assumed that prokaryotes whose ploidy had been reported in previous studies were monoploid. This limitation may pose little problem because bacteria are generally assumed to be monoploid'. This comment says it all really. I can only think that the authors did not bother to read any of our papers on this matter, as our whole argument is based on extreme ploidy in giant bacteria. How can this study claim to be a reanalysis when it does not even begin to ask the same questions?

Our Response:

Sorry. This is a mistake. We assumed that prokaryotes whose ploidy had ****not**** been reported in any previous studies were monoploid. We considered ploidy level in data analyses and obtained similar conclusion.

Reviewer's Comment:

I honestly came to this paper with the intent to read it carefully and hopefully learn something; and if the analysis challenged what I had thought to be true, then I should modify my opinion. I hope that this review makes it abundantly clear why I do not think this analysis challenges in any meaningful way the Lane-Martin hypothesis. I cannot recommend the publication of this paper in its current form.

Yours sincerely

Nick Lane

Our Response:

Thanks to your kind advices and suggestions, the quality of this study largely increased. We believe that the revised manuscript meets the criteria for publication in Royal Society Open Science.

Appendix B

=====
Response to Editor (Prof Dr Rees Kassen)
=====

Editor's Comment:

First off, I would like to thank the authors for their patience regarding the evaluation of this manuscript. As explained in my original letter to the authors, the preliminary views by reviewers were so divided that re-review would likely be necessary. This has now happened, with three new reviewers providing comments.

On the basis of these reviews I am recommending the paper be accepted for publication following minor revisions. As you will see two of the reviewers are quite favourable but one is not. The critical reviewer suggests that there is not sufficient advance in this paper over previous work, since the data largely recapitulate previous findings. I disagree. I think the expanded data set, the addition of a phylogenetically-informed analysis, and the revised analyses that include exponential growth rates (rather than just minimal growth rates) provides a more powerful test of the energetic complexity hypothesis than has previously been available. This is a significant advance, given the limited data that has previously been provided to test the hypothesis.

That said I would encourage the authors to tone down some of their claims about lack of support for the energetic hypothesis and in favour of the non-adaptive hypothesis. The data provided do not, in fact, provide direct evidence for the latter; indeed, this wasn't the point of the paper. As for the argument that there is no evidence for an energetic barrier, well, I would say there is less compelling evidence than has previously been claimed but that the jury is still out on how and why complexity evolved in eukaryotes and not prokaryotes. Reality may be more complicated than either camp would like to think, as hinted at by Reviewer 2.

Our Response:

We really appreciate your kind handling of our manuscript. This study could not be completed without you (and also the reviewers, you selected). We were really happy you are the editor of our manuscript. We carefully revised the manuscript again according to your and reviewers' comments. Finally, we believe that our manuscript meets the criteria of publication.

Editor's Comment:

Specific suggestions for improvement:

1. I would revise this sentence in the abstract: 'Our findings indicate that there is no energetic barrier to genome complexity between prokaryotes and eukaryotes, suggesting that the prokaryote-eukaryote divide is inexplicable from the energetic perspective' to be less strident about the energetic barrier. There could still be a barrier, although it may have been overcome in different ways by prokaryotes and eukaryotes. The difference in complexity is not 'inexplicable', just harder to explain.

Our Response:

We avoid to mention about the energetic barrier and revised as follows: our findings indicate that the prokaryote-eukaryote divide is hard-to-explain from the energetic perspective.

Editor's Comment:

2. line 55: the hypothesis is not 'debatable'. It is 'controversial'.

Our Response:

Corrected.

Editor's Comment:

3.Line 274 - see comments on the abstract above. There could still be a barrier, and mitochondria could still play a role in overcoming that barrier, but the results suggest the actual evolutionary pathway to complexity was itself a more complex process than either hypothesis would have it.

Our Response:

As above, we avoid to mention about the energetic barrier. We simply mentioned as follows: the findings indicate the prokaryote-eukaryote divide is harder-to-explain than previously thought.

Editor's Comment:

4. Finally, Reviewer 3 has some suggestions for improving the text that are valuable as well. Please revise accordingly.

Our Response:

We revised the manuscript according to the reviewers' comments.

=====
Response to Reviewer 3
=====

Reviewer's Comment:

Chiyomaru and Takemoto (C&T) revisit the energetics of genome complexity by gathering more data. In the foreground of the file on this paper stands a debate between two L&M teams Lynch&Marionov 2015 (LM15) and Lane&Martin 2010 (LM10). The peer review on the paper has been intense with R1 for the Lynch side talking about "nails in coffins", "nonsensical", a "bizzare rant" versus R2 for the Lane side pointing out in very level headed words that the T&C paper has some issues and that the Lynch side still does not get the point.

The favourable referee R1 weighing in for Lynch says that C&T is "redundant" with earlier work, and that the case for publication is weak ("a difficult call") and that the conclusions are "muddled" because C&T cannot find that L&M10 are really wrong in any way, findings that R1 requests to be restated so as to support a different conclusions. The critical referee R2 has identified fundamental problems with the paper that revisions and new analyses have not resolved.

There has been enough written on this paper, a decision is needed. Referees are supposed to advise Editors. My recommendation to the editor

is to decline. The reason is that this referee looked at the RSOS guidelines which state:

Our Response:

We would thank you for taking the time to review our manuscript. Our manuscript was modified according to your comments. We summarized our responses at the bottom of your comments.

Reviewer's Comment:

Papers should only be recommended for publication if they meet these [five] criteria:

[1] - Original research which has not been published elsewhere.

[2] - Submissions should sufficiently advance scientific knowledge. Negative findings, meta-analyses and studies testing reproducibility of significant work are also encouraged. Repeated experiments will only be considered if they provide a meaningful contribution to the literature. Derivative work will not be considered.

[3] - Conclusions are supported by the data; data supporting the findings of the paper are publicly available.

[4] - Compliance with appropriate ethical standards.

[5] - Experimental protocols/procedure and statistical analysis are performed to a high technical standard and are both methodologically and scientifically sound.

Criterion [1] appears to be met.

Criterion [2], the referees' long comments and the debate between referees in the present file show how incremental the advance is, if there is one. C&T find that L&M10 is reproducible, the first 2 paragraphs of the results state this clearly. Then comes a discussion about cell mass, which is volume because unicellular forms cells have roughly the same density and are 80% water. It is known that some prokaryotes are large, as both L&Ms point out. On lines 200-214 T&C report analyses indicating the eukaryote cells are larger than prokaryote cells (l. 206); that is decidedly not an advance. On lines 215-229 T&C then show that cell mass (volume) also has a phylogenetic component. The result is Table 2 where the reader is challenged to find a strong statistical signal -- the values are all very similar.

Criterion [3], the conclusions of L&M10 are supported by the present data, as R1 recurrently points out asking that it not be mentioned, not the conclusions of C&T that L&M15 are supported. The data appears to be available.

Criterion [4], compliance with appropriate ethical standards, there are some accusations about data availability in the text that are hardly acceptable for a scientific paper, in my view. That is not good from an ethics standpoint. The issue on the debate ref 37 vs 38 is 37 having been rejected (37 states so in the text) by the journal that published ref. 11.

Criterion [5], this referee is not an expert in statistics.

The discussion starts out by saying that (1.231) "These results indicate no difference in power per genome and power per gene between prokaryotes and eukaryotes, which is not consistent with Lane and Martin's conclusion that the prokaryotic genome size is constrained by bioenergetics". From there the discussion gets very, very dense, missing the bigger picture presented in L&M10. I think the broad appeal of L&M10 is to the community of physiologists and cell biologists. Eukaryotes respire in mitochondria, prokaryotes respire at the plasma membrane, eukaryotes are complex, prokaryotes are not, eukaryotes have meiosis, prokaryotes do not.

On the bottom line, as this reader/referee sees it, C&T say that mitochondria had no role in evolution. L&M10 are saying that symbiosis might be an important force in evolution that can create novelty, while L&M15 are saying that that all evolution needs in order to create novelty is variation in population size. C&T report data that R1 finds redundant and R2 finds to be entirely missing the point.

I think that R2 has the better arguments, hence my recommendation above. I really like symbiosis in evolution, hence I like L&M10. Martin published a good paper entitled "Symbiogenesis, gradualism and mitochondrial energy in eukaryote evolution" on this debate in an obscure but readily accessible open access journal. He pointed out that mitochondria produce about 95% of the ATP in a respiring cell, but comprise only 10% of the cell's volume, releasing constraints on the remaining 90% of the cell (mostly cytosol) to evolve other functions, structural proteins rather than enzymes, while leaving everything under that control of a handful of more genes whose expression not copy number is increased. How can that be irrelevant for evolution? Discussing that would help to explain what the actual issue is here, because it was the role of mitochondria in the origin of complexity of eukaryotic cells that was at the heart of the L&M10 paper. Especially the Discussion is not focussed on science, it is about who did what to whom, which is immaterial for understanding the role of mitochondria in eukaryote origin. The "debate" as presented in the C&T paper is hardly enlightening, as both R1 and R2 conclude, albeit for very different reasons.

Our Response:

Please remember that we never think mitochondria are irrelevant for evolution. As mention in the manuscript, we believe that the origin of the mitochondrion was a key event in evolutionary history. Our study is limited to the energetics of genome complexity. We also think that mitochondria play different important roles in evolution. We cited your paper entitled "Symbiogenesis, gradualism..." and clearly mentioned that mitochondria played important roles in evolutionary history to acquire eukaryote-specific traits such as the cell cycle, sex, phagocytosis, endomembrane trafficking, the nucleus and multicellularity. Moreover, we still believe that our study is enlightening. As the editor also mentioned, the expanded data set, the addition of a phylogenetically-informed analysis, and the revised analyses provides a more powerful test of the energetic complexity hypothesis than has previously been available. We believe that this is a significant advance, given the limited data that has previously been provided to test the hypothesis.

=====
Response to Reviewer 4
=====

Reviewer's Comment:

I've checked all the boxes because they are required, but I really have not had the time to reason my way through the arguments here, in which I am not in any case an expert. My concerns about Lane and Martin 2010 are nevertheless emphasized by authors' Fig. 1, which shows overlap between prokaryotes and eukaryotes, however measured. This seems to me to make the only point that needs to be made: we might easily imagine a large-genomed, high-energy prokaryote "evolving into" a small-genomed, low-energy eukaryote. Indeed that's what most sensible folks would imagine - not that something like the average modern prokaryote "evolved into" something like the average modern eukaryote. Thus comparisons of averages seem to me beside the point. Moreover, there are eukaryotes without mitochondria that have most of the bells and whistles of other eukaryotes without the mitochondrion-derived energy. So such a scenario is possible.

Our Response:

We would thank you for taking the time to review our manuscript. We really happy you get straight to the point what we want to say. We completely agree with your comments.

=====
Response to Reviewer 5
=====

Reviewer's Comment:

I feel for the authors getting "crushed" between the strongly opposing (even acrimonious) viewpoints of both original reviewers. Although I do not agree with the overall vision of reviewer 1 (e.g. stating "...but the final discussion in this paper is a bit like leaving room for the possibility for intelligent design or for disagreement on global warming." I consider completely overblown) he/she makes some important points. I looked at how the authors dealt with previous criticisms of the two reviewers and think the response is adequate.

I think this revision is acceptable for publication, though I have the feeling that we are in the middle of a muddled discussion and the authors missed a chance of contributing more insight (e.g. what is going on with the two prokaryotic outliers at the top in figures 1C/1D?).

Our Response:

We would thank you for taking the time to review our manuscript. We agree with you. Moreover, we are happy you give a positive comment. The outliers are Thioploca and Trichodesmium. We also mentioned these in the manuscript. Specifically, we mentioned that the cell mass and, power per genome, and power per gene of several prokaryotes (Thioploca and Trichodesmium species) almost equaled to those of eukaryotes.

Reviewer's Comment:

I have some problems with this part of the discussion (though the authors do not have to change anything on my account):

Our Response:

We carefully revise the manuscript according to your comments.

Reviewer's Comment:

Line 271-276: "The findings eliminate the need to invoke an energetics barrier hypothesis to genome complexity between prokaryotes and eukaryotes; rather, they support the hypothesis of the passive emergence of genome complexity by non-adaptive processes [9,11,12]. As Lynch and Marinov [12] mentioned, the origin of the mitochondrion was not a prerequisite for genome-size expansion, although the origin was a key event in evolutionary history;..." This part and the rest of the paragraph seems to want to convey that cell mass has nothing to do with ATP generation/consumption and that the co-occurrence of the endosymbiont acquisition and the increase of cellular (including genome) complexity is just a coincidence. In what way is it a key event then??? These kind of passages remind me that though Lane and Martin may be wrong about some of the details, overall I still prefer their explanations to the alternatives.

Our Response:

Yes, we strongly recognize the importance of mitochondria. We clearly mentioned that mitochondria played important roles in evolutionary history to acquire eukaryote-specific traits such as the cell cycle, sex, phagocytosis, endomembrane trafficking, the nucleus and multicellularity.

Reviewer's Comment:

Minor comments (mostly stylistic/language errors).

Line 39: "However, not all prokaryotes have evolved biological complexity." I would say: "However, only some prokaryotes have evolved biological complexity."

Our Response:

Corrected.

Reviewer's Comment:

Line 281: "The definition of power per genome and power per gene is still controvertible." INCORRECT Language use. "The definition of power per genome and power per gene is still a matter of controversy."

Our Response:

Corrected.